# Ablation of palladin in adult heart causes dilated cardiomyopathy associated with intercalated disc abnormalities

Giuseppina Mastrototaro[1,2†], Pierluigi Carullo[2,3], Jianlin Zhang[4], Beatrice Scellini[5], Nicoletta Piroddi[5], Simona Nemska[2], Maria Carmela Filomena[2,3], Simone Serio[2], Carol A Otey[6], Chiara Tesi[5], Fabian Emrich[7], Wolfgang A Linke[8], Corrado Poggesi[5], Simona Boncompagni[9], Marie-Louise Bang[2,3]*

[1]University of Milan-Bicocca, Milan, Italy; [2]IRCCS Humanitas Research Hospital, Milan, Italy; [3]Institute of Genetic and Biomedical Research, National Research Council, Milan unit, Milan, Italy; [4]University of California, San Diego, La Jolla, United States; [5]Department of Experimental and Clinical Medicine, University of Florence, Florence, Italy; [6]Department of Cell Biology and Physiology, University of North Carolina at Chapel Hill, Chapel Hill, United States; [7]Department of Cardiac Surgery, Goethe University, Frankfurt, Germany; [8]Institute of Physiology II, University of Münster, Münster, Germany; [9]Department of Neuroscience, Imaging and Clinical Sciences and Center for Advanced Studies and Technology, University G. d'Annunzio, Chieti, Italy

**\*For correspondence:**
marie-louise.bang@cnr.it

**Present address:** [†]AGC Biologics, Milan, Italy

**Competing interest:** The authors declare that no competing interests exist.

**Abstract** Palladin (PALLD) belongs to the PALLD/myopalladin (MYPN)/myotilin family of actin-associated immunoglobulin-containing proteins in the sarcomeric Z-line. PALLD is ubiquitously expressed in several isoforms, and its longest 200 kDa isoform, predominantly expressed in striated muscle, shows high structural homology to MYPN. *MYPN* gene mutations are associated with human cardiomyopathies, whereas the role of PALLD in the heart has remained unknown, partly due to embryonic lethality of PALLD knockout mice. In a yeast two-hybrid screening, CARP/Ankrd1 and FHOD1 were identified as novel interaction partners of PALLD's N-terminal region. To study the role of PALLD in the heart, we generated conditional (cPKO) and inducible (cPKOi) cardiomyocyte-specific PALLD knockout mice. While cPKO mice exhibited no pathological phenotype, ablation of PALLD in adult cPKOi mice caused progressive cardiac dilation and systolic dysfunction, associated with reduced cardiomyocyte contractility, intercalated disc abnormalities, and fibrosis, demonstrating that PALLD is essential for normal cardiac function. Double cPKO and MYPN knockout (MKO) mice exhibited a similar phenotype as MKO mice, suggesting that MYPN does not compensate for the loss of PALLD in cPKO mice. Altered transcript levels of *MYPN* and *PALLD* isoforms were found in myocardial tissue from human dilated and ischemic cardiomyopathy patients, whereas their protein expression levels were unaltered.

## Editor's evaluation

This valuable work will be of interest to scientists who study cardiomyocyte homeostasis and contraction. Using solid methodology, this study is the first to assess the consequences of cardiomyocyte-specific knockout of Palladin, identifying a compensation mechanism that takes place when this gene is deleted in embryogenesis, but not in adulthood. In addition, this study identifies novel Palladin interactors, suggests a role for Palladin in the maintenance of intercalated disc structure, and assesses levels of Palladin expression in human samples.

## Introduction

Palladin (PALLD) is an actin-associated protein, which, together with myopalladin (MYPN) and myotilin (MYOT), belongs to a small protein family of immunoglobulin (Ig) domain-containing proteins in the Z-line associated with the actin cytoskeleton (reviewed in *Otey et al., 2009*). While MYPN and MYOT are specifically expressed in striated and skeletal muscle, respectively, PALLD is ubiquitously expressed in various tissues, including the heart, where it is expressed at high levels (*Wang and Moser, 2008*). Mutations in the *MYPN* gene have been associated with dilated (DCM), hypertrophic (HCM), and restrictive cardiomyopathy (*Bagnall et al., 2010*; *Duboscq-Bidot et al., 2008*; *Meyer et al., 2013*; *Purevjav et al., 2012*), while *MYOT* gene mutations can cause various skeletal muscle disorders (reviewed in *Otey et al., 2009*). Mutations in the *PALLD* gene have been linked to pancreatic cancer (*Liotta et al., 2021*; *Pogue-Geile et al., 2006*; *Slater et al., 2007*), and PALLD levels have been shown to correlate with increased invasiveness of cancer cells (*Gilam et al., 2016*; *Goicoechea et al., 2009*; *von Nandelstadh et al., 2014*). However, the role of PALLD in the heart has remained unknown. Whereas MYPN and MYOT are expressed as single isoforms, PALLD exists in many different isoforms due to the presence of four alternative promoters, alternative splicing, and early termination as illustrated in *Figure 1—figure supplement 1* (*Goicoechea et al., 2010*; *Wang and Moser, 2008*). The longest 200 kDa isoform of PALLD is predominantly expressed in striated muscle and highly homologous in structure to MYPN, sharing five Ig domains and a central proline-rich region (*Otey et al., 2009*). Furthermore, PALLD contains an additional proline-rich region, not shared with MYPN. The 140 kDa and 90 kDa isoforms containing four and three Ig domains, respectively, are also expressed in the heart, but have a more ubiquitous expression pattern (*Wang and Moser, 2008*). Molecular weights and expression profiles of the other isoforms have not been determined in detail (*Goicoechea et al., 2010*; *Wang and Moser, 2008*). Within the cardiac Z-line, PALLD and MYPN share several interaction partners, including α-actinin (*Bang et al., 2001*; *Rönty et al., 2004*), nebulette (*Bang et al., 2001*), titin (*Filomena et al., 2021*), and members of the PDZ-LIM protein family (cypher/ZASP/LDB3, ALP/PDLIM3, CLP36/PDLIM1, RIL/PDLIM4) (*von Nandelstadh et al., 2009*) as illustrated in *Figure 1A*. Furthermore, PALLD and MYPN bind to myocardin-related transcription factors (MRTFs) (*Filomena et al., 2020*; *Jin et al., 2010*) as well as bind and bundle F-actin (*Dixon et al., 2008*; *Filomena et al., 2020*; *Gurung et al., 2016*), stabilizing the actin cytoskeleton and consequently promoting serum response factor (SRF) signaling. Additionally, PALLD binds to sorbin and SH3 domain-containing 2 (SORBS2) (*Rönty et al., 2005*), localized at the intercalated disc (ICD) and Z-line of the heart, as well as various proteins expressed in non-muscle cells, including actin-associated (*Boukhelifa et al., 2006*; *Boukhelifa et al., 2004*; *Goicoechea et al., 2006*; *Jin et al., 2007*; *Mykkänen et al., 2001*; *Rachlin and Otey, 2006*) and signaling/adaptor (*Lai et al., 2006*; *Rönty et al., 2007*) proteins.

Consistent with the interaction of PALLD with F-actin and numerous actin binding proteins, PALLD is associated with various actin-based structures, including focal adhesions, stress fibers, cell-cell junctions, and Z-lines, and is important for the assembly, organization, and maintenance of the actin cytoskeleton (reviewed in *Jin, 2011*; *Otey et al., 2009*). PALLD is well-known for its involvement in cell motility and smooth muscle cell differentiation, implicating it in cancer (*Jin, 2011*), while its role in the heart has remained elusive, in part due to embryonic lethality of PALLD knockout mice (*Luo et al., 2005*). Thus, to provide insights into the specific role of PALLD in cardiac muscle, we generated conditional (cPKO) and inducible (cPKOi) cardiomyocyte (CMC)-specific knockout mice for the most common PALLD isoforms. While conditional knockout of PALLD in the heart did not cause any pathological cardiac phenotype, PALLD knockdown in adult heart resulted in left ventricular (LV) dilation and systolic dysfunction, associated with fibrosis and ICD abnormalities. Furthermore, we identified cardiac ankyrin repeat protein (CARP/Ankrd1) and formin homology 2 domain containing 1 (FHOD1) as novel interaction partners of the N-terminal region of PALLD. Quantitative real-time PCR (qRT-PCR) analyses on myocardial tissue from DCM and ischemic cardiomyopathy (ICM) patients showed increased transcript levels of *MYPN* and the *PALLD* long isoform in DCM patients, while total transcript levels of *PALLD* were reduced in both DCM and ICM patients. On the other hand, no changes in the protein levels of MYPN and PALLD were found.

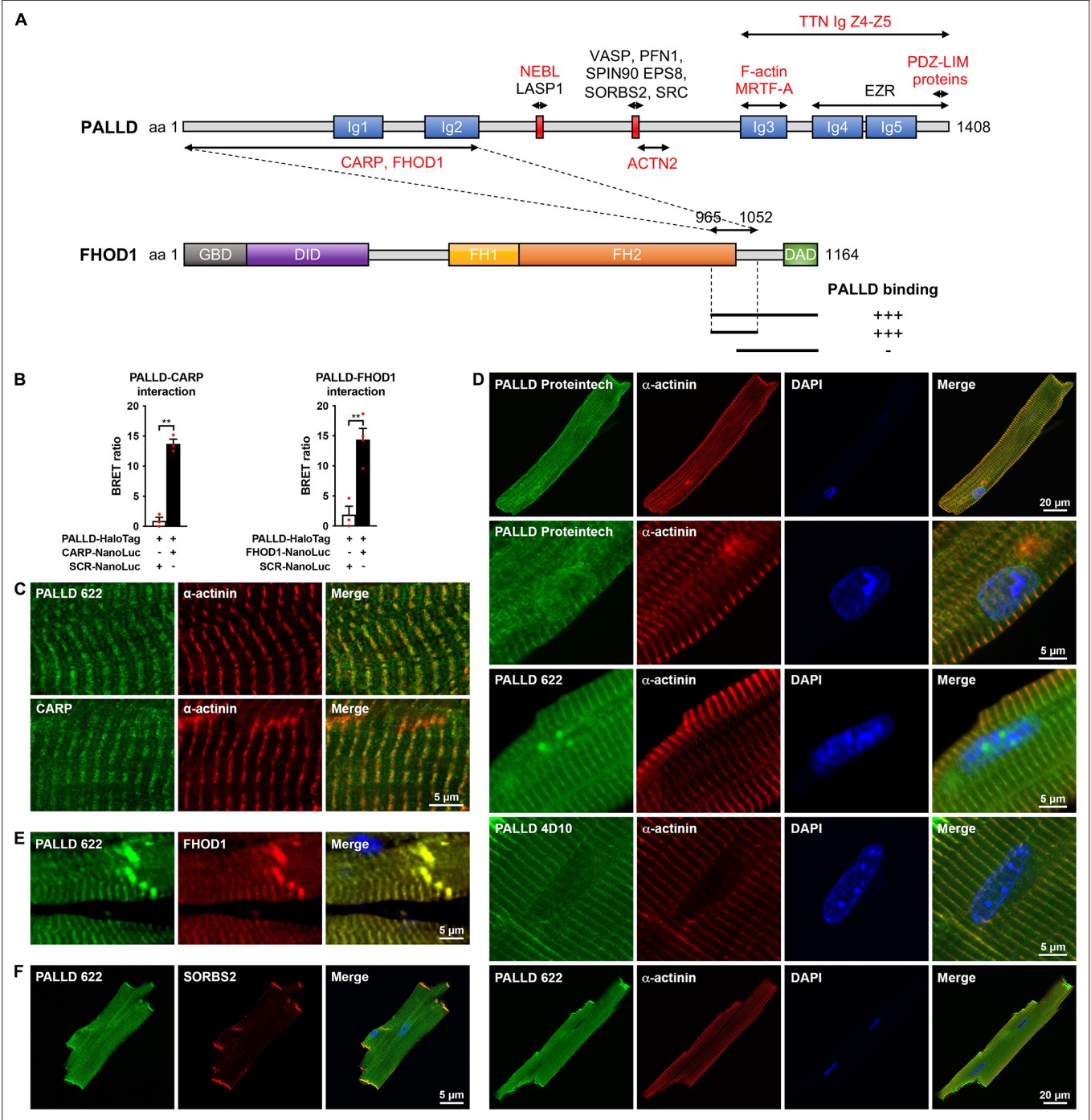

**Figure 1.** Palladin (PALLD) binds to cardiac ankyrin repeat protein (CARP/Ankrd1) and formin homology 2 domain containing 1 (FHOD1) and is localized in the Z-line, I-band, intercalated disc (ICD), and nucleus in cardiomyocytes (CMCs). (**A**) Schematic representation of the domain structure of PALLD. Immunoglobulin (Ig) domains are shown in blue and proline-rich regions are indicated in red. Binding sites for known and novel interaction partners are shown with the ones in common with myopalladin (MYPN) in red. In a yeast two-hybrid screening, the N-terminal region of PALLD, comprising two Ig domains, was found to bind to CARP and FHOD1, as illustrated. The interacting region in FHOD1 was narrowed down to a region within the C-terminal region of FHOD1 (residue 965–1052), as indicated. GBD, GTPase-binding domain; DID, diaphanous inhibitory domain; FH1, formin homology 1 domain; FH2, formin homology 2 domain; DAD, diaphanous autoregulatory domain. (**B**) Confirmation of PALLD-CARP and PALLD-FHOD1 interactions in NanoBRET protein interaction assays with CARP-NanoLuc or FHOD1-NanoLuc as donor and fluorescently labeled PALLD-HaloTag as acceptor. Data are represented as mean ± standard error of the mean (SEM) (n = 3–4). \*\*p<0.01; Student's *t*-test. (**C**) Immunofluorescence analysis of sectioned heart by STED microscopy, showing localization of both PALLD and CARP in the sarcomeric I-band. (**D**) Confocal fluorescence microscopy showing the presence of PALLD in the nucleus of CMCs. Both the PALLD 622 antibody and the PALLD Proteintech antibody showed nuclear localization of PALLD in most

*Figure 1 continued on next page*

*Figure 1 continued*

CMCs (an example of a CMC without nuclear staining for PALLD is shown on the bottom), while the PALLD 4D10 antibody did not stain the nucleus. Nuclei are visualized by DAPI (blue). (**E**) Confocal fluorescence microscopy of sectioned heart, showing colocalization of PALLD and FHOD1 at the ICD and the Z-line. (**F**) Confocal fluorescence microscopy showing colocalization of PALLD and Sorbin and SH3 domain-containing 2 (SORBS2) at the ICD in CMCs.

The online version of this article includes the following source data and figure supplement(s) for figure 1:

**Source data 1.** Confirmation of PALLD-CARP and PALLD-FHOD1 interactions in NanoBRET protein interaction assays.

**Figure supplement 1.** Schematic representation of known murine PALLD protein isoforms and their domain organization.

**Figure supplement 2.** Schematic representation of the murine *Palld* gene structure with known transcripts indicated below.

**Figure supplement 3.** Yeast-two hybrid (Y2H) assays showing the interaction of the N-terminal region of palladin (PALLD) with cardiac ankyrin-repeat protein (CARP/Ankrd1) and formin homology 2 domain containing 1 (FHOD1).

**Figure supplement 4.** Confocal fluorescence microscopy showing the presence of palladin (PALLD) in the nucleus of cardiomyocytes both from wild-type (WT) mice and cardiac ankyrin repeat protein (CARP/Ankrd1) knockout (CKO) mice.

## Results

### The PALLD N-terminal region binds to CARP/Ankrd1 and FHOD1

To identify potential binding partners to the N-terminal region of the longest 200 kDa PALLD isoform, of which no interaction partners have previously been identified, we performed a yeast two-hybrid screening of a human heart cDNA library using the N-terminal region of human PALLD, comprising the Ig1 and Ig2 domains, as bait. Among the potential interacting proteins were cardiac ankyrin repeat protein (CARP/Ankrd1) and formin homology 2 domain containing 1 (FHOD1) (*Figure 1A* and *Figure 1—figure supplement 3*).

The interaction of PALLD with CARP, a transcriptional cofactor and known interaction-partner of the corresponding region of MYPN (*Bang et al., 2001*; *Miller et al., 2003*), was confirmed in a NanoBRET protein interaction assay (*Figure 1B*) and stimulated emission depletion (STED) immunofluorescence microscopy showed the localization of both CARP and PALLD in the I-band of the sarcomere (*Figure 1C*). Furthermore, PALLD was frequently found in the nucleus of CMC, where also CARP is known to localize (*Figure 1D*; *Bang et al., 2001*; *Miller et al., 2003*). Interestingly, two distinct polyclonal antibodies recognizing all PALLD isoforms (PALLD 622) (*Goicoechea et al., 2010*; *Pogue-Geile et al., 2006*) and the PALLD C-terminal region (Proteintech Group), respectively, showed localization of PALLD in the nucleus, while a monoclonal antibody (4D10) specific for the second proline-rich region (*Parast and Otey, 2000*) did not stain the nucleus. PALLD was found to be absent from nuclear regions strongly stained with DAPI, corresponding to heterochromatin, suggesting the localization of PALLD in the transcriptionally active euchromatin regions of the nucleus. Additionally, stronger PALLD staining in a punctuate pattern was often observed in regions with absent DAPI staining, likely corresponding to nucleoli. Immunofluorescence stainings on CMCs from CARP knockout mice (*Bang et al., 2014*) showed no effect on the nuclear localization of PALLD, leaving out the possibility that CARP is responsible for the targeting of PALLD to the nucleus (*Figure 1—figure supplement 4*). We did not attempt to further narrow down the interaction site in CARP as we previously demonstrated that full-length CARP is required for binding to MYPN (*Bang et al., 2001*).

FHOD1 belongs to the formin protein family involved in the regulation of the actin cytoskeleton (*Al Haj et al., 2015*; *Dwyer et al., 2014*). In the yeast two-hybrid screening, we identified two clones containing the 200 C-terminal amino acid residues of FHOD1 (*Figure 1A*). By cotransformation of PALLD with truncated FHOD1 constructs in yeast, the PALLD interaction site was further narrowed down to a region comprising the C-terminal part of the formin homology 2 (FH2) domain and part of a unique sequence before the C-terminal diaphanous autoregulatory domain (DAD) (residue 965–1052; *Figure 1A* and *Figure 1—figure supplement 3B*). The FH2 domain is required for self-association of FHOD1, essential for its actin regulating activity (*Madrid et al., 2005*; *Takeya and Sumimoto, 2003*). The N-terminal region of MYPN likewise bound to FHOD1 (*Figure 1—figure supplement 3B*). The PALLD-FHOD1 interaction was confirmed in a NanoBRET protein interaction assay (*Figure 1B*) and confocal microscopy showed colocalization of PALLD with FHOD1 at the ICD as well as the Z-line where FHOD1 is present at lower levels (*Figure 1E*). At the ICD, PALLD also colocalized with its known binding partner SORBS2 (*Figure 1F*).

## Conditional CMC-specific PALLD knockout mice (cPKO) exhibit normal cardiac morphology and function

To overcome the embryonic lethality of PALLD-deficient mice (*Luo et al., 2005*), which has previously precluded the analysis of the role of PALLD in the heart, we generated *Palld* floxed (*Palld*<sup>fl/fl</sup>) mice containing LoxP sites flanking exon 14, present in the major isoforms (*Figure 1—figure supplement 2* and *Figure 2—figure supplement 1*). *Palld*<sup>fl/fl</sup> mice were crossed with*Myh6*-nuclear-Cre mice (*Abel et al., 1999*) to generate conditional CMC-specific PALLD knockout (cPKO) mice in which PALLD is ablated in developing CMCs from around embryonic day 11 (E11) (*Chen et al., 2006*). Efficient knockdown of the most common PALLD isoforms (200 kDa, 140 kDa, 90 kDa isoforms) in the heart was confirmed by qRT-PCR (*Figure 2A*), showing an ~75% knockdown in CMCs based on levels of transcripts encoding the cardiac-specific 200 kDa PALLD isoform, which is not expressed in other cardiac cell types, such as fibroblasts and smooth muscle cells. Consistently, western blot analysis showed nearly complete knockout of PALLD at the protein level (*Figure 2B*) with an ~95% reduction of the highest expressed 90 kDa isoform (*Figure 2C*). The expression of a smaller weakly expressed PALLD isoform of about 70 kDa was not affected. This isoform likely corresponds to a short N- or C-terminal isoform not including exon 14 (see *Figure 1—figure supplement 2*) and was previously detected also in human pancreas (*Goicoechea et al., 2010*).

The effect of PALLD knockout on cardiac morphology and function was analyzed by echocardiography. As shown in *Figure 2—figure supplement 2AFigure 2—figure supplement 2—source data 1*, no significant differences were found between cPKO and control male mice in any of the parameters analyzed up to 6 mo of age. Consistently, histological analyses showed no cardiac alterations in cPKO mice compared to control mice (*Figure 2—figure supplement 2B and C*). Furthermore, in response to mechanical pressure overload induced by transaortic constriction (TAC), no significant differences were found between cPKO and *Myh6*<sup>Cre/+</sup> control mice as shown by echocardiography (*Figure 2D* and *Figure 2—source data 3*) and histological analyses (*Figure 2E–G*).

## Double knockout mice for PALLD and MYPN are indistinguishable from MYPN knockout mice

Due to the high structural homology between PALLD and MYPN, we hypothesized that the essentially absent pathological cardiac phenotype of cPKO mice is due to compensation by MYPN. To test this, we generated double knockout mice for PALLD and MYPN (cPKO/MKO dKO). We recently reported that MYPN knockout (MKO) mice develop LV dilation and systolic dysfunction starting from 4 mo of age (*Filomena et al., 2021*). By echocardiography, cPKO/MKO dKO (*Mypn*<sup>-/-</sup>/*Palld*<sup>fl/fl</sup>;*Myh6*<sup>Cre/+</sup>) mice were found to be indistinguishable from MKO mice both from the MKO line and derived from the crosses as *Mypn*<sup>-/-</sup>/*Palld*<sup>fl/fl</sup> littermates (*Figure 2H* and *Figure 2—source data 6*), indicating that MYPN is unlikely to compensate for the loss of PALLD in cPKO mice.

## Ablation of PALLD in adult mice results in DCM and systolic dysfunction

Hypothesizing that compensatory mechanisms during development prevent the development of a pathological phenotype of cPKO mice, we investigated the effect of PALLD knockout in adult CMCs. *Palld*<sup>fl/fl</sup> mice were crossed with *Myh6*-MerCreMer transgenic mice (*Sohal et al., 2001*) to generate inducible CMC-specific PALLD knockout (cPKOi) mice, which were induced by tamoxifen (TAM) injection at 7 wk of age. Effective knockdown of the most common PALLD isoforms in the heart was confirmed by qRT-PCR 8 wk after induction, showing an about 80% reduction in transcripts encoding the cardiac-specific 200 kDa PALLD isoform (*Figure 3A*). Consistently, western blot analysis on isolated CMCs showed efficient knockout of the targeted isoforms (*Figure 3B*) with an ~84% knockdown of the highest expressed 90 kDa isoform (*Figure 3C*). Like in cPKO mice, the antibody also recognized a protein of around 70 kDa present at similar levels in both cPKOi mice and control mice.

Echocardiography showed the development of LV dilation and progressive systolic dysfunction in cPKOi male mice starting from 8 wk after induction (*Figure 3D and E*, and *Figure 3—source data 3*). Intraventricular septum thickness was significantly increased 24 wk after TAM injection, whereas LV posterior wall thickness and heart weight to body weight were unaltered. To rule out the possibility that the phenotype of cPKOi mice is due to the toxic effect of TAM, we injected cPKO mice with TAM and evaluated cardiac function by echocardiography 8 wk after TAM injection (*Figure 3—figure supplement 1* and *Figure 3—figure supplement 1—source data 1*). No pathological cardiac

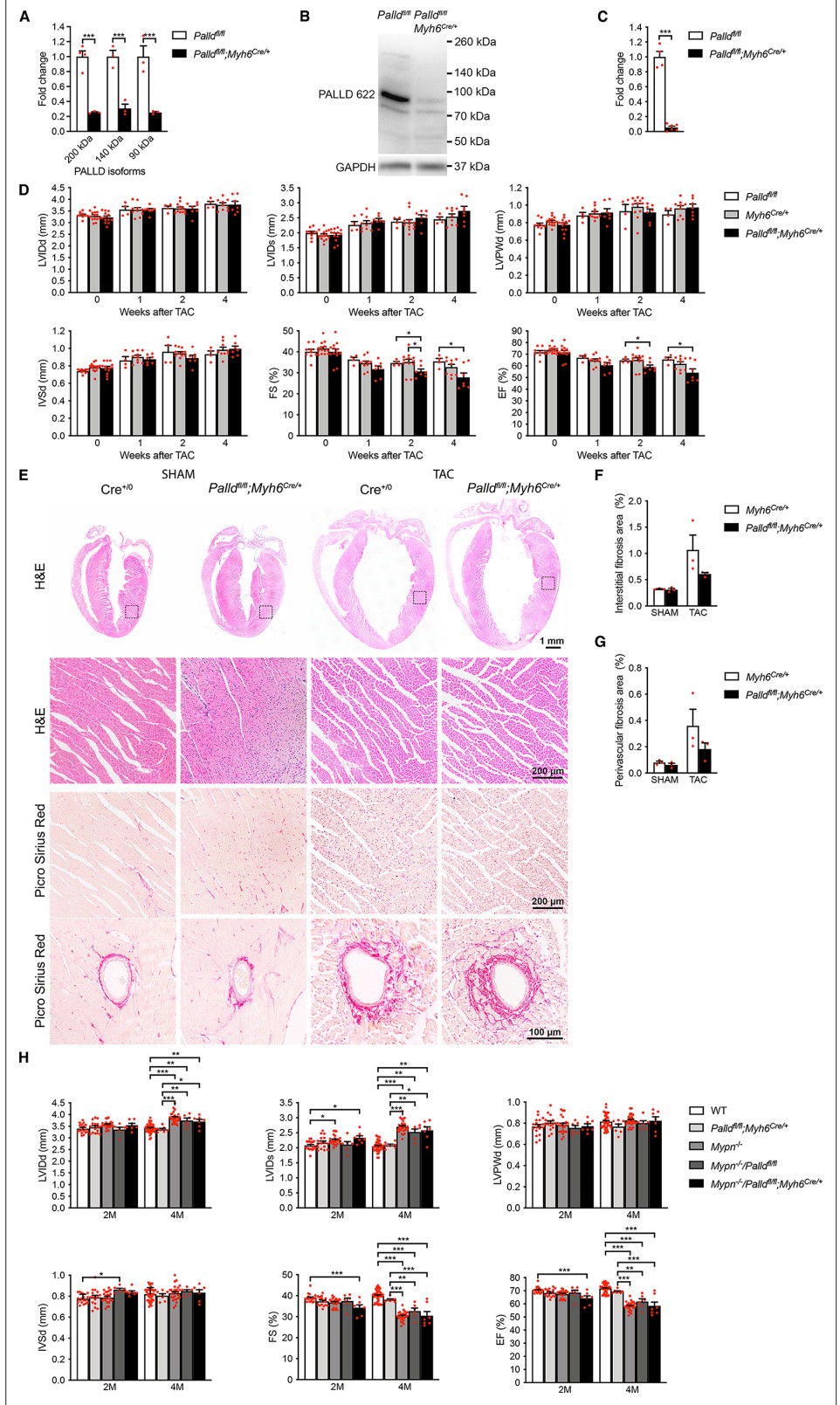

**Figure 2.** Echocardiographic and histological analyses of cardiomyocyte-specific palladin (cPKO) mice as well as cPKO and myopalladin (MYPN) double knockout (cPKO/MKO dKO) mice. (**A**) Quantitative real-time PCR (qRT-PCR) for transcripts encoding the most common PALLD isoform on left ventricular (LV) RNA from 2-mo-old cPKO (*Palld*^fl/fl;*Myh6*^Cre/+) and control (*Palld*^fl/fl) male mice (n = 3–4 biological replicates and three technical replicates per

*Figure 2 continued on next page*

*Figure 2 continued*

group). Data were normalized to *Gapdh*. \*\*\*p<0.001; two-way analysis of variance (ANOVA) with Šidák's multiple comparison test. (**B**) Western blot analysis for PALLD (PALLD 622 antibody) on LV lysate from cPKO and control male mice. (**C**) Densitometric analysis for the 90 kDa PALLD isoform using GAPDH for normalization (n = 4–6 per group). \*\*\*p<0.001; Student's *t*-test. (**D**) Echocardiography analyses of cPKO and control male mice under basal conditions and 1, 2, and 4 wk after transaortic constriction (TAC). Pressure gradient >70 mmHg. LVID, left ventricular inner diameter; LVPW, left ventricular posterior wall thickness; IVS, interventricular septum thickness; FS, fractional shortening; EF, ejection fraction; d, diastole; s, systole (n = 4–14 per group). \*p<0.05; \*\*p<0.01; linear mixed model with Tukey's multiple comparisons test. (**E**) Hematoxylin and eosin (H&E) and Picro Sirius Red stainings of heart sections from cPKO and control male mice 4 wk after TAC or SHAM. The areas indicated with a box are shown at higher magnification below. Examples of vessels are shown on the bottom. (**F, G**) Percent area of interstitial fibrosis (**F**) and perivascular fibrosis (**G**) in the LV (n = 3 per group). No statistical difference by one-way ANOVA with Tukey's multiple comparisons test. (**H**) Echocardiographic analysis of cPKO/MKO dKO vs. single KO and control male mice at 2 and 4 mo (**M**) of age. See (**D**) for abbreviations (n = 6–42 per group). \*p<0.05; \*\*p<0.01; \*\*\*p<0.001; linear mixed model with Tukey's multiple comparisons test. All data are represented as mean ± standard error of the mean (SEM).

The online version of this article includes the following source data and figure supplement(s) for figure 2:

**Source data 1.** Quantitative real-time PCR (qRT-PCR) and densitometry of western blots on cardiomyocyte-specific palladin (cPKO) and control male mice.

**Source data 2.** Uncropped western blots for *Figure 2B*.

**Source data 3.** Echocardiographic parameters of 2-mo-old cardiomyocyte-specific palladin knockout (cPKO) male mice compared to controls before and 4 wk after mechanical pressure overload induced by transaortic constriction (TAC).

**Source data 4.** Echocardiographic analysis on cardiomyocyte-specific palladin (cPKO) and control male mice subjected to transaortic constriction (TAC) or SHAM.

**Source data 5.** Quantification of interstitial and vascular fibrosis in the left ventricle of cardiomyocyte-specific palladin (cPKO) and control male mice subjected to transaortic constriction (TAC) or SHAM.

**Source data 6.** Echocardiographic parameters of 3- and 6-mo-old cardiomyocyte-specific palladin and myopalladin double knockout (cPKO/MKO dKO) mice subjected to transaortic constriction (TAC) or SHAM.

**Source data 7.** Echocardiographic analysis on 2- and 4-mo-old cardiomyocyte-specific palladin (cPKO) and myopalladin (MYPN) double knockout and control male mice.

**Figure supplement 1.** Generation of *Palld* floxed mice.

**Figure supplement 2.** Echocardiographic and histological analyses of cardiomyocyte-specific palladin knockout (cPKO) male mice.

**Figure supplement 2—source data 1.** Echocardiographic parameters of 3- and 6-mo-old cardiomyocyte-specific palladin knockout (cPKO) male mice compared to controls under basal conditions.

**Figure supplement 2—source data 2.** Echocardiographic analysis on 3- and 6-mo-old cardiomyocyte-specific palladin (cPKO) and control male mice.

**Figure supplement 2—source data 3.** Quantification of interstitial and vascular fibrosis in the left ventricle of 6-mo-old cardiomyocyte-specific palladin (cPKO) and control male mice.

---

phenotype was observed after TAM injection, suggesting that early ablation of PALLD allows for the activation of compensatory mechanisms. Consistent with the reduced systolic function of cPKOi mice, analysis of contractile function and twitch $Ca^{2+}$ transients in ventricular CMCs 12 wk after TAM induction showed reduced sarcomere shortening and $Ca^{2+}$ transient amplitude in cPKOi mice compared to control mice (*Figure 3F*). Time to peak and time to relaxation as well as the rise and decay of $Ca^{2+}$ transients were unaffected. Similarly, measurements of biomechanical properties of myofibril preparations from the LV, showed reduced maximum $Ca^{2+}$-activated isometric tension ($P_0$) of cPKOi myofibrils compared to controls 8 wk after TAM induction (see *Table 1*). The kinetics of force generation and relaxation following rapid $Ca^{2+}$ increase and removal were similar in cPKOi and control mice, indicating that the impairment of active force generation in cPKOi mice is not due to a change in the number of force generating cross-bridges. The resting tension at optimal sarcomere length between cPKOi and control preparations was not significantly different between cPKOi and control preparations (*Table 1*) and the sarcomere length-tension relationship was similar in cPKOi and control preparations, although

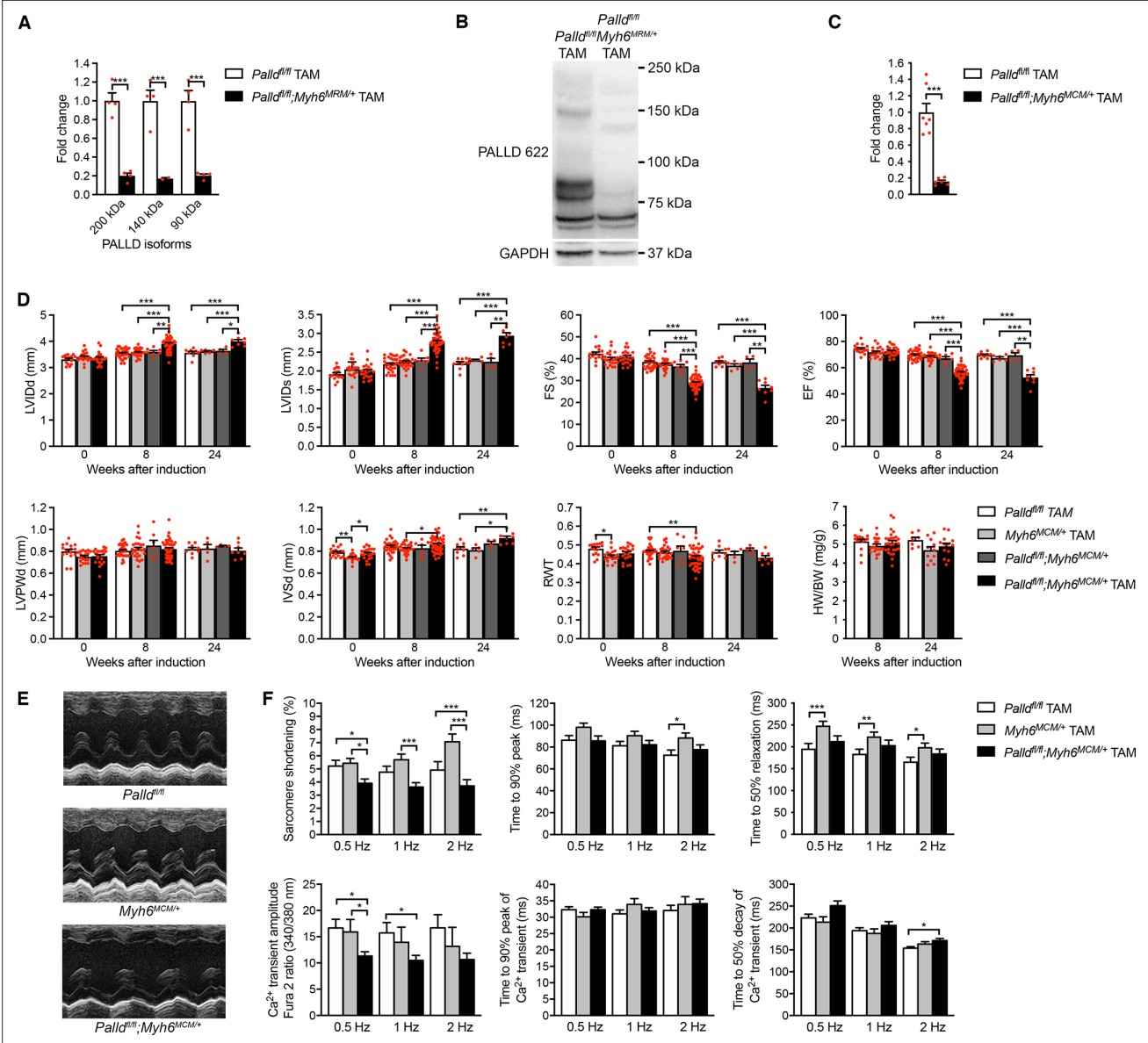

**Figure 3.** Echocardiographic and functional analyses of inducible cardiomyocyte-specific palladin knockout (cPKOi) mice. (**A**) Quantitative real-time PCR (qRT-PCR) for transcripts encoding the most common palladin (PALLD) isoforms on left ventricular (LV) RNA from cPKOi (*Palld^fl/fl*;*Myh6^MCM/+* TAM) and control (*Palld^fl/fl* TAM) mice 8 wk after tamoxifen (TAM) induction (n = 4 biological replicates and three technical replicates per group). Data were normalized to *Gapdh*. ***p<0.001; two-way analysis of variance (ANOVA) with Šidák's multiple comparisons test. (**B**) Western blot analysis for PALLD (PALLD 622 antibody) on adult cardiomyocytes (CMCs) from cPKOi and control mice 8 wk after TAM induction. (**C**) Densitometric analysis for the 90 kDa PALLD isoform using GAPDH for normalization (n = 6–7). ***p<0.001; Student's *t*-test. (**D**) Echocardiographic analysis of cPKOi and control male mice 0, 8, and 24 wk after TAM induction. See (**D**) for abbreviations. RWT, relative wall thickness ((LVPWd + IVSd)/LVIDd); HW, heart weight, BW, body weight (n = 4–44 per group). *p<0.05; **p<0.01; ***p<0.001; linear mixed model with Tukey's multiple comparisons test. (**E**) Representative echocardiographic short-axis M-mode images of hearts from cPKOi and control mice 24 wk after TAM induction. (**F**) IonOptix analysis of ventricular CMC contractility and Ca^2+ transients in cPKOi male mice vs. control mice 12 wk after TAM induction. Top, Sarcomere shortening, time to 90% peak, and time to 50% relaxation (n = 33 cells from 4 *Palld^fl/fl* TAM mice, n = 34 cells from five MCM^+/0 TAM mice, and n = 52 cells from six *Palld^fl/fl*;*Myh6^MCM/+* TAM mice). Bottom, amplitude of Ca^2+ transient, time to 90% peak of Ca^2+ transient, and time to 50% decay of Ca^2+ transient (n = 39 cells from four *Palld^fl/fl* TAM mice, n = 18 cells from three MCM^+/0 TAM mice, and n = 58 cells from six *Palld^fl/fl*;*Myh6^MCM/+* TAM mice). *p<0.05; **p<0.01; ***p<0.001; two-level hierarchical testing with Bonferroni correction (*Sikkel et al., 2017*). All data are represented as mean ± standard error of the mean (SEM).

The online version of this article includes the following source data and figure supplement(s) for figure 3:

**Source data 1.** Quantitative real-time PCR (qRT-PCR) and densitometry of western blots on inducible cardiomyocyte-specific palladin (cPKOi) and control male mice.

*Figure 3 continued on next page*

*Figure 3 continued*

**Source data 2.** Uncropped western blots for *Figure 3B*.

**Source data 3.** Echocardiographic parameters of inducible cardiomyocyte-specific palladin knockout (cPKOi) male mice compared to controls 8 and 24 wk after tamoxifen (TAM) induction.

**Source data 4.** Echocardiographic analysis on inducible cardiomyocyte-specific palladin (cPKOi) and control male mice 8 and 24 wk after tamoxifen (TAM) induction.

**Source data 5.** Heart weight to body weight ratio (HW/BW) measurements on inducible cardiomyocyte-specific palladin (cPKOi) and control male mice 8 and 24 wk after tamoxifen (TAM) induction.

**Source data 6.** Analysis of sarcomere shortening and $Ca^{2+}$ transients in cardiomyocytes from inducible cardiomyocyte-specific palladin (cPKOi) and control male mice 12 wk after tamoxifen (TAM) induction.

**Figure supplement 1.** Echocardiographic analysis of cardiomyocyte-specific palladin (cPKO) and control male mice 8 wk after tamoxifen (TAM) injection.

**Figure supplement 1—source data 1.** Echocardiographic parameters of cardiomyocyte-specific palladin knockout (cPKO) male mice compared to controls 8 weeks after tamoxifen (TAM) injection.

**Figure supplement 1—source data 2.** Echocardiographic analysis on cardiomyocyte-specific palladin knockout (cPKO) and control male mice 8 wk after tamoxifen (TAM) injection.

**Figure supplement 2.** Sarcomere-length tension relationship in cardiac myofibrils from the left ventricle of inducible cardiomyocyte-specific palladin knockout (cPKOi) and control male mice 8 wk after tamoxifen (TAM) induction.

**Figure supplement 2—source data 1.** Data for sarcomere-length tension relationship in cardiac myofibrils from the left ventricle of inducible cardiomyocyte-specific palladin knockout (cPKOi) and control male mice 8 wk after tamoxifen (TAM) induction.

there was a tendency towards lower resting tension at longer sarcomere lengths (*Figure 3—figure supplement 2*).

Histological analyses 24 wk after TAM induction showed significantly increased interstitial and perivascular fibrosis in the heart of cPKOi mice compared to control hearts (*Figure 4A and B*). No apoptosis was detected by TUNEL staining at 4 and 24 wk after TAM induction. Analysis of isolated CMCs showed increased CMC size in cPKOi mice compared to control mice 12 wk after TAM induction (*Figure 4C*), which was mainly due to increased CMC width (*Figure 4D and E*).

## Upregulation of markers of pathological cardiac remodeling in cPKOi mouse heart

To determine the effect of PALLD knockout in adult mice on cardiac gene expression, qRT-PCR for various genes involved in cardiac remodeling, fibrosis, apoptosis, and inflammation was performed on LV RNA 24 wk after TAM induction (*Figure 4—source data 4*). This revealed upregulation of the cardiac stress responsive genes *Nppb*, *Myh7*, and *Ankrd1* (*Figure 4F* and *Figure 4—source data 4*), while *Nppa*, *Myh6*, *Acta1*, and *Actc1* were unaffected. In addition, although our histological analyses showed interstitial fibrosis in cPKOi hearts, no significant changes were found in the expression levels

**Table 1.** Tension generation and relaxation in ventricular myofibrils from inducible cardiomyocyte-specific palladin knockout (cPKOi) mice and control male mice 8 wk after tamoxifen (TAM) induction.

| | Resting conditions | | Tension generation | | Relaxation | | |
| | | | | | Slow phase | | Fast phase |
| Myofibril batch | SL (µm) | RT (mN mm$^{-2}$) | P$_0$ (mN mm$^{-2}$) | $k_{ACT}$ (s$^{-1}$) | Duration (ms) | $k_{REL}$ (s$^{-1}$) | $k_{REL}$ (s$^{-1}$) |
|---|---|---|---|---|---|---|---|
| *Palld*$^{fl/fl}$ TAM | 2.13 ± 0.02 (30) | 8.65 ± 1.03 (30) | 138 ± 7 (30) | 4.98 ± 0.30 (30) | 78.2 ± 3.9 (25) | 2.02 ± 0.22 (25) | 20.8 ± 2.2 (29) |
| *Myh6*$^{MCM/+}$ TAM | 2.13 ± 0.02 (33) | 9.01 ± 1.02 (33) | 129 ± 9 (33) | 4.91 ± 0.28 (31) | 74.7 ± 3.5 (27) | 1.54 ± 0.19 (27) | 28.4 ± 2.8 (27) |
| *Palld*$^{fl/fl}$;*Myh6*$^{MCM/+}$ TAM | 2.14 ± 0.02 (23) | 9.67 ± 1.81 (23) | 111 ± 8 (23) | 4.83 ± 0.43 (23) | 86.5 ± 4.8 (22) | 1.86 ± 0.19 (22) | 21.5 ± 1.2 (22) |

All data are represented as mean ± standard error of the mean (SEM). Numbers in parentheses are number of myofibrils. SL, sarcomere length, RT, resting tension, P$_0$, maximum isometric tension; $k_{ACT}$, rate constant of tension rise following step-wise pCa decrease (8.0→4.5) by fast solution switching. Full tension relaxation kinetics were characterized by the duration and rate constant of tension decay of the isometric slow relaxation phase (slow $k_{REL}$) and the rate constant of the fast relaxation phase (fast $k_{REL}$).; one-way analysis of variance (ANOVA) with Šidák's multiple comparisons test. *$p < 0.05$ vs. *Palld*$^{fl/fl}$ TAM.

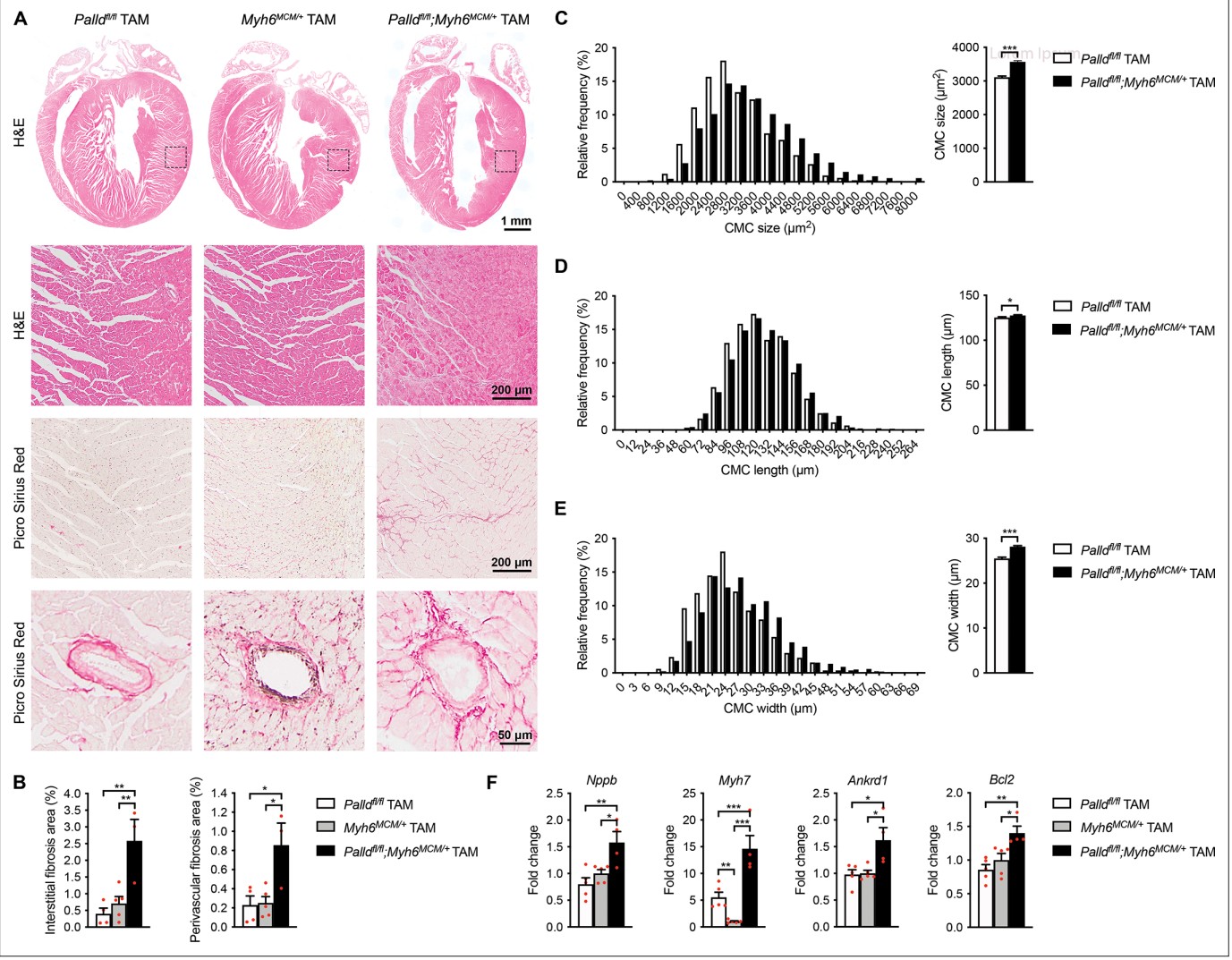

**Figure 4.** Histological and molecular analyses of cardiomyocyte-specific palladin knockout (cPKOi) mice. (**A**) Hematoxylin and eosin (H&E) and Picro Sirius Red stainings of heart sections from cPKOi and control male mice 24 wk after tamoxifen (TAM) induction. The areas indicated with a box are shown at higher magnification. Examples of vessels are shown on the bottom. (**B**) Percent area of interstitial fibrosis (left) and perivascular fibrosis (right) in the left ventricle (LV) (n = 3–5 per group). **p<0.01; one-way analysis of variance (ANOVA) with Tukey's multiple comparisons test. (**C–E**) Left, frequency distributions of cardiomyocyte (CMC) size (n = 830 CMCs from three *Palld*$^{fl/fl}$ mice and 1079 CMCs from three *Palld*$^{fl/fl}$;*Myh6*$^{MCM/+}$ TAM mice) (**C**), length (n = 877 CMCs from three *Palld*$^{fl/fl}$ mice and 1576 CMCs from three *Palld*$^{fl/fl}$;*Myh6*$^{MCM/+}$ TAM mice) (**D**), and width (n = 842 CMCs from three *Palld*$^{fl/fl}$ mice and 1565 CMCs from three *Palld*$^{fl/fl}$;*Myh6*$^{MCM/+}$ TAM mice) (**E**) in cPKOi and control male mice 12 wk after TAM induction. Right, average CMC size (**C**), length (**D**), and width (**E**) in the same mice. *p<0.05; ***p<0.001; Student's t-test. (**F**) Quantitative real-time PCR (qRT-PCR) analysis on LV RNA from cPKOi and control mice 24 wk after TAM induction (n = 4–5 biological replicates and three technical replicates per group). Data were normalized to *Gapdh*. *p<0.05; **p<0.01; ***p<0.001; one-way ANOVA with Tukey's multiple comparisons test. All data are represented as mean ± standard error of the mean (SEM).

The online version of this article includes the following source data for figure 4:

**Source data 1.** Cardiomyocyte (CMC) size, length, and width measurements in inducible cardiomyocyte-specific palladin (cPKOi) and control male mice 12 wk after tamoxifen (TAM) induction.

**Source data 2.** Quantification of interstitial and perivascular fibrosis in the left ventricle of inducible cardiomyocyte-specific palladin (cPKOi) and control male mice 24 wk after tamoxifen (TAM) induction.

**Source data 3.** Raw data for quantitative real-time PCR (qRT-PCR) on inducible cardiomyocyte-specific palladin (cPKOi) and control male mice 24 wk after tamoxifen (TAM) induction.

**Source data 4.** Quantitative real-time PCR (qRT-PCR) analysis on left ventricle from inducible cardiomyocyte-specific palladin knockout (cPKOi) male mice compared to controls 24 wk after tamoxifen (TAM) induction.

of *Col1a2*, *Col3a1*, *Ctgf*, *Tgfb1*, and *Acta2*. Consistent with the absence of apoptosis in cPKOi hearts, the antiapoptotic gene *Bcl2* was upregulated (**Figure 4F**) and the apoptosis-related genes *Bax* and *Trp53* were unaffected. No changes were found in genes encoding PALLD interacting proteins (*Nebl*, *Ldb3/Zasp/Cypher*, *Pdlim3*) or other sarcomeric proteins (*Des* and *Tnnc1*). The upregulation of *Nppb*, *Myh7*, and *Ankrd1* suggests cardiac remodeling and the presence of chronic stress in the heart of cPKOi mice, consistent with an essential role of PALLD for normal heart function.

## Intercalated disc abnormalities in cPKOi mouse hearts

To determine the ultrastructural effects of PALLD knockout, transmission electron microscopy (TEM) analyses were performed on papillary muscle from cPKOi and control male mice 8 and 24 wk after TAM induction. The most noticeable observation was alterations in ICD ultrastructure in cPKOi mice both 8 and 24 wk after TAM induction (**Figure 5**). In particular, cPKOi mice showed ICDs with abnormal ultrastructure, characterized by partial or total disruption of the ordered folded regions at the adherens junction, often showing an evidently widened amplitude (**Figure 5A**). Quantification of ICD fold amplitude (i.e., the lateral width of the folded regions indicated in **Figure 5B**) revealed a higher, much more variable ICD fold amplitude in cPKOi compared to control mice as shown by the frequency distribution plots in **Figure 5C and D** (left), showing a rightward shift in cPKOi mice both at 8 and 24 wk after TAM induction, with the greatest variability observed 8 wk after TAM induction. Consequently, the average ICD fold amplitude was increased by ~98.7% and ~76.6% in cPKOi mice vs. control mice 8 and 24 wk after TAM induction, respectively (**Figure 5C and D**, right). Detailed analysis of regions containing high-amplitude ICDs showed very convoluted and abnormal fold regions, often occupied by thick filaments constituting the A-band of the sarcomere with an evident disruption of the thin actin-containing filament areas normally observed within the folds of control ICDs (**Figure 5E**). Another frequent observation was the presence of structural abnormalities in the regions adjoining high-amplitude ICDs (**Figure 5F**). Specifically, while most sarcomeres had a normal organization, areas with wavy Z-lines and sarcomere disruption were present in the proximity of high-amplitude ICDs, where accumulation of vesicles was also often observed in areas of degeneration.

Based on the location of PALLD at the ICD and the ICD abnormalities observed in cPKOi mice, we sought to examine whether PALLD ablation affects the localization or expression of other ICD proteins, including PALLD's interaction partners SORBS2 and FHOD1 as well as components of the desmosome (desmoplakin, plakoglobin/γ-catenin, plakophilin-2, desmocollin-2, desmoglein-2, desmin), adherens junction (N-cadherin, α-E-catenin, β-catenin, plakoglobin/γ-catenin, vinculin), and gap junction (connexin 43). Immunofluorescence stainings of isolated adult CMCs showed no alterations in the localization of ICD proteins (**Figure 6—figure supplement 1**) and PALLD interacting proteins, including CARP, FHOD1, nebulette, SORBS2, MRTF-1, and cypher (**Figure 6—figure supplement 2**). However, western blot analyses showed an ~1.5-fold increase in SORBS2 expression in the LV of cPKOi mice 8 wk after TAM induction (**Figure 6A and B**). No other changes in the expression of ICD proteins were observed. Furthermore, no changes in the levels of the PALLD homologue MYPN and PALLD-associated proteins were found (**Figure 6—figure supplement 3**). Since we recently found that PALLD binds to titin Ig-domains Z4-Z5 in the Z-line (**Filomena et al., 2021**) as well as indirectly to the titin N2A region via CARP (**Bang et al., 2001**; **Miller et al., 2003**), we also determined titin isoform expression and phosphorylation, both known to affect passive stiffness (**Loescher et al., 2022**), but found no changes between cPKOi and control mice (**Figure 6—figure supplement 4**), consistent with absent effect of PALLD KO on resting tension (**Table 1** and **Figure 3—figure supplement 2**). Furthermore, determination of total and phosphorylation levels of various proteins involved in cardiac signaling showed no alterations (**Figure 6—figure supplement 3**), suggesting that the DCM phenotype of cPKOi mice is caused by structural changes in the ICD.

## Altered transcript levels of *PALLD* and *MYPN* in myocardial biopsies from human DCM and ICM patients

Our findings demonstrating an important role of PALLD for normal cardiac function prompted us to determine whether transcript levels of *PALLD* and its striated muscle-specific homologue *MYPN* are altered during ischemic and nonischemic cardiac disease. qRT-PCR analysis on LV biopsies from DCM and ischemic cardiomyopathy (ICM) patients vs. healthy control hearts (for patient details, see **Figure 6—source data 6**) revealed an ~2.9-fold upregulation of *MYPN* mRNA in DCM patients as

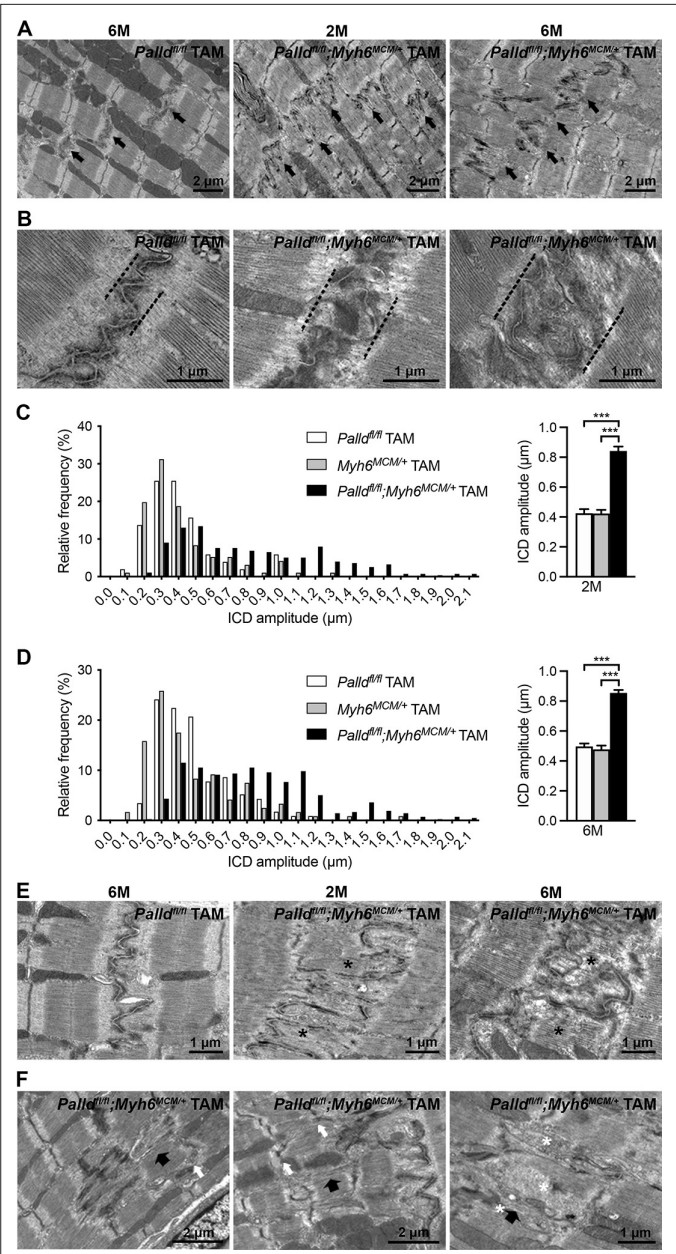

**Figure 5.** Transmission electron microscopy (TEM) analysis of papillary muscles from inducible cardiomyocyte-specific palladin knockout (cPKOi) and control mice. (**A**) Representative electron micrographs of papillary muscles from control (*Palld*^fl/fl TAM) and cPKOi (*Palld*^fl/fl;*Myh6*^MCM/+ TAM) male mice 8 and 24 wk after tamoxifen (TAM) induction. Black arrows indicate intercalated discs (ICDs). (**B**) Representative electron micrographs of ICD ultrastructure from 6-mo-old control and cPKOi mice. Dashed black lines indicate ICD regions. (**C, D**) ICD fold amplitude frequency distribution (left) and average (right) in control (*Palld*^fl/fl TAM, *Myh6*^MCM/+ TAM) and cPKOi samples 2 mo (n = 96 measurements on 13 ICDs from 1 *Palld*^fl/fl mouse, n = 51 measurements on 7 ICDs from one *Myh6*^MCM/+ TAM mouse, and 279 measurements on 33 ICDs from 2 *Palld*^fl/fl;*Myh6*^MCM/+ TAM mice) (**C**) and 6 mo (n = 96 measurements on 15 ICDs from 1 *Palld*^fl/fl mice, n = 116 measurements on 15 ICDs from 1 MCM^+/0 TAM mouse, and 420 measurements on 48 ICDs from 2 *Palld*^fl/fl;*Myh6*^MCM/+ TAM mice) (**D**) after TAM induction. Data are represented as mean ± standard error of the mean (SEM). ***p<0.001; Brown–Forsythe and Welch's ANOVA with Games–Howell's multiple comparisons test. (**E**) Representative electron micrographs of ICDs from control and cPKOi mice 8 and 24 wk after TAM induction. Asterisks show thick filaments in abnormal fold regions. (**F**) Electron micrographs showing altered sarcomeric regions (large black arrows) in the proximity of high-amplitude ICDs in papillary muscles from 6-mo-old cPKOi mice. White arrows point to jagged Z-lines. Asterisks highlight groups of vesicles.

*Figure 5 continued on next page*

*Figure 5 continued*

The online version of this article includes the following source data for figure 5:

**Source data 1.** Intercalated disc (ICD) fold amplitude measurements in inducible cardiomyocyte-specific palladin (cPKOi) and control male mice 8 and 24 wk after tamoxifen (TAM) induction.

well as an ~1.9-fold upregulation of *PALLD* mRNA encoding the 200 kDa striated muscle-specific PALLD isoform, structurally homologous to MYPN (*Figure 6B*). In contrast, the overall transcript level of all PALLD isoforms except for truncated N-terminal transcript(s) was ~2.8- and ~3.0-fold reduced in DCM and ICM patients, respectively. As previously reported, *ANKRD1* was upregulated in both DCM patients (~6.2-fold) and ICM patients (~3.3-fold) (*Bogomolovas et al., 2015*; *Herrer et al., 2014*; *Kempton et al., 2018*; *Nagueh et al., 2004*; *Zolk et al., 2002*). Western blot analyses showed no differences in the expression of MYPN and PALLD 90 kDa and 200 kDa isoforms at the protein level, while CARP/Ankrd1 was 2.3-fold upregulated (*Figure 6D and E*).

## Discussion

In this study, we show that in addition to its known localization in the sarcomeric Z-line, PALLD is located in the I-band, the ICD, and the nucleus of CMCs. The PALLD C-terminal region was previously found to target to the nucleus in podocytes (*Endlich et al., 2009*) and smooth muscle cells (SMCs) (*Jin et al., 2010*), while the N-terminal region was shown to be required for nuclear export (*Endlich et al., 2009*). Consistently, immunofluorescence stainings with antibodies towards all PALLD isoforms or the PALLD C-terminal region showed the localization of PALLD in the nucleus of CMCs. However, surprisingly, antibodies against the second proline-rich region present in the major isoforms did not stain the nucleus. This suggests that only isoforms without the second proline-rich region (indicated in *Figure 1—figure supplement 1*) are present in the nucleus, which is consistent with immunohisto-chemical stainings of human biopsies from pancreas (*Goicoechea et al., 2010*). Thus, PALLD isoforms may have different cellular functions depending on the presence or absence of the second proline-rich region. As shown in *Figure 1A*, many interaction partners of PALLD bind to its second proline-rich region, including SORBS2, which is present in the ICD and Z-line of the heart, so one possibility could be that the second proline-rich region tethers PALLD to the actin cytoskeleton, preventing it from going to the nucleus.

In a yeast two-hybrid screening, we identified CARP and FHOD1 as novel interaction partners of the PALLD N-terminal region for which no interaction partners have previously been identified. Since CARP is a well-known interaction partner of the MYPN N-terminal region (*Bang et al., 2001*), struc-turally homologous to the PALLD N-terminal region, it is not surprising that also PALLD binds to CARP. CARP and MYPN are known to form a complex in the I-band, where CARP is linked to the titin N2A region (*Bang et al., 2001*; *Miller et al., 2003*). Consistently, we found that also PALLD is located in the I-band but hypothesize that only the 200 kDa PALLD isoform containing the N-terminal CARP-binding region is targeted to the I-band. Titin is a largest known protein, stretching from the Z-line to the M-line and functioning as a molecular spring responsible for the passive stiffness of striated muscle, implicating it in force transmission and mechanosensing (*Loescher et al., 2022*). Thus, based on their link to titin and dual localization in the sarcomere and the nucleus, CARP and MYPN were previously proposed to form a complex involved in the transduction of stretch-induced signaling from the sarco-mere to the nucleus (*Laure et al., 2010*; *Miller et al., 2003*). To determine whether CARP may be responsible for the targeting of PALLD to the nucleus, we performed immunofluorescence stainings of CMCs from CARP knockout mice. However, CARP ablation did not affect the localization of PALLD in the nucleus of CMCs, questioning the hypothesis of a role of PALLD and MYPN in mechanosensing. Consistent with targeting experiments in podocytes and SMCs, it is therefore likely that the PALLD C-terminal region is responsible for its targeting to the nucleus also in CMCs. In particular, based on the binding of the PALLD Ig3 domain to the transcriptional cofactor MRTF-A (*Filomena et al., 2020*; *Jin et al., 2010*), which cycles between the cytosol and nucleus to regulate SRF signaling, it is tempting to speculate that MRTF-A is responsible for the targeting of PALLD to the nucleus. Another possibility could be that PALLD contains a nuclear localization signal (NLS), as previously predicted by sequence analysis (*Endlich et al., 2009*). On the other hand, no NLSs were predicted in MYPN, which is also present in the nucleus. Based on the high homology between the two proteins, we therefore

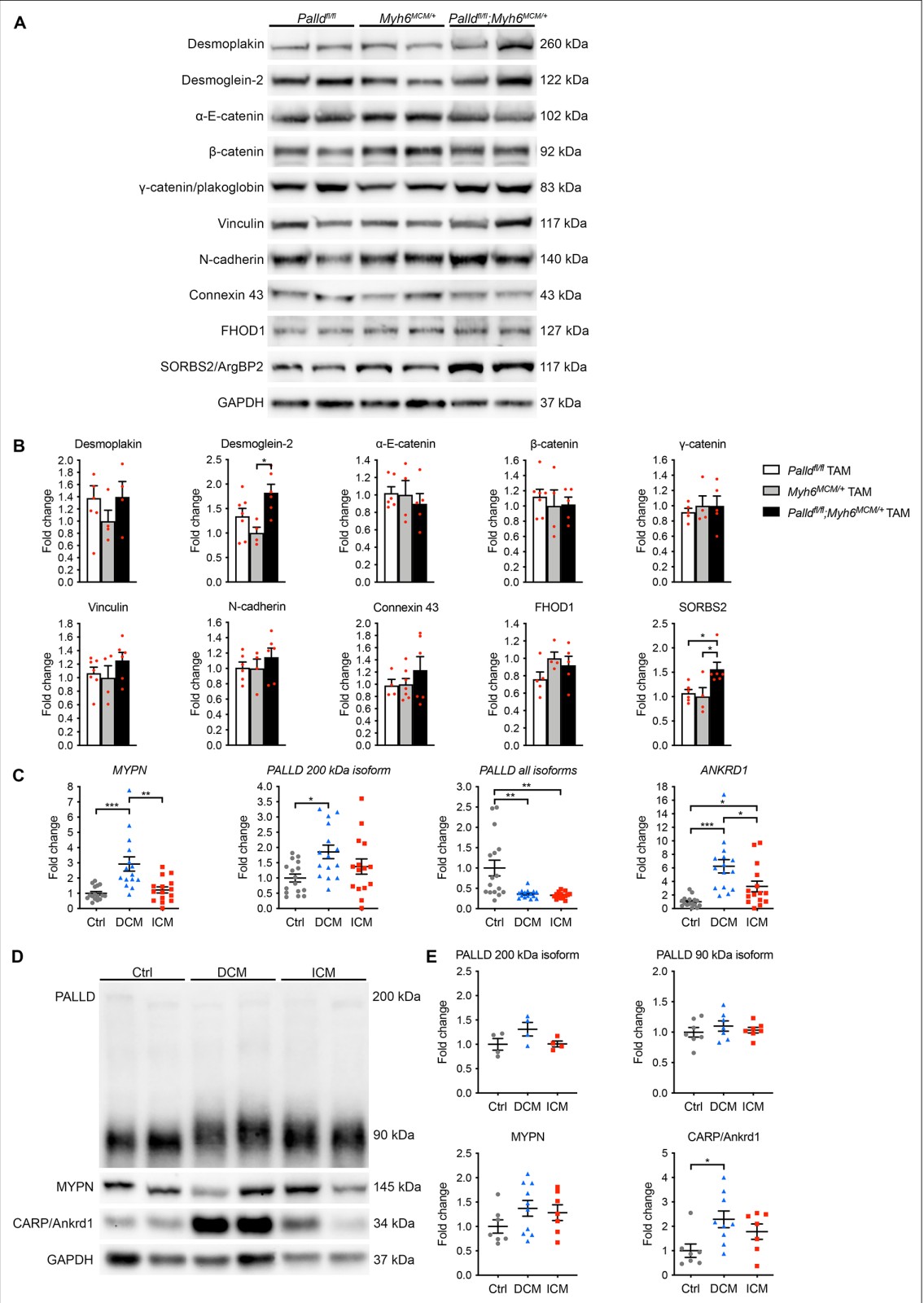

**Figure 6.** Molecular analyses on inducible cardiomyocyte-specific palladin knockout (cPKOi) mice and human cardiomyopathy patients. (**A**) Western blot analyses for intercalated disc (ICD) proteins on left ventricular (LV) lysate from cPKOi mice compared to control male mice 8 wk after TAM induction. Representative blots are shown. GAPDH was used as loading control. (**B**) Densitometric analysis of blots in (**A**) using GAPDH for normalization (n = 3–4 per group). *p<0.05, one-way analysis of variance (ANOVA) with Tukey's multiple comparisons test. (**C**) Quantitative real-time PCR (qRT-PCR) analysis for

*Figure 6 continued on next page*

*Figure 6 continued*

*MYPN*, *PALLD*, and *ANKRD1* on LV biopsies from dilated cardiomyopathy (DCM) (n = 15) and ischemic cardiomyopathy (ICM) (n = 15) male patients vs. healthy control (Ctrl) hearts (n = 16). Data were normalized to *PGK1*. \*p<0.05, \*\*p<0.01; Brown–Forsythe and Welch's ANOVA with Dunnett's T3 multiple comparisons test. (D) Western blot analysis for PALLD (PALLD 622 antibody), MYPN, and CARP/Ankrd1 on lysate from LV biopsies of DCM and DCM male patients vs. healthy Ctrl hearts. (E) Densitometric analyses using total protein content as assessed on TGX Stain-Free gels (Bio-Rad Laboratories) for normalization (n = 4–10 per group). \*p<0.05; Brown–Forsythe and Welch's ANOVA with Dunnett's T3 multiple comparisons test. All data are represented as mean ± standard error of the mean (SEM).

The online version of this article includes the following source data and figure supplement(s) for figure 6:

**Source data 1.** Uncropped western blots for *Figure 6A*.

**Source data 2.** Densitometry of western blots on inducible cardiomyocyte-specific palladin (cPKOi) and control male mice 8 wk after tamoxifen (TAM) induction.

**Source data 3.** Quantitative real-time PCR (qRT-PCR) analysis on left ventricular biopsies from patients with dilated (DCM) and ischemic (ICM) cardiomyopathy vs. healthy control (Ctrl) hearts.

**Source data 4.** Uncropped western blots for *Figure 6D*.

**Source data 5.** Densitometry of western blots on left ventricular biopsies from patients with dilated (DCM) and ischemic (ICM) cardiomyopathy vs. healthy control (Ctrl) hearts.

**Source data 6.** Patient characteristics.

**Figure supplement 1.** Immunofluorescence stainings for intercalated disc proteins of cardiomyocytes from inducible cardiomyocyte-specific palladin knockout (cPKOi) and control male mice 16 wk after tamoxifen (TAM) induction.

**Figure supplement 2.** Immunofluorescence stainings of cardiomyocytes for palladin (PALLD) interacting proteins from inducible cardiomyocyte-specific palladin knockout (cPKOi) and control male mice 16 wk after tamoxifen (TAM) induction.

**Figure supplement 3.** Western blot analysis for palladin (PALLD) interacting proteins and proteins involved in cardiac signaling pathways on left ventricular lysate from inducible cardiomyocyte-specific palladin knockout (cPKOi) and control male mice 8 wk after tamoxifen (TAM) induction.

**Figure supplement 3—source data 1.** Uncropped western blots.

**Figure supplement 4.** Titin isoform expression and phosphorylation in the left ventricular of inducible cardiomyocyte-specific palladin knockout (cPKOi) and control male mice 8 weeks after tamoxifen (TAM) induction.

**Figure supplement 4—source data 1.** Uncropped western blots.

consider it most likely that PALLD and MYPN are transported to the nucleus through binding to proteins that translocate to the nucleus, such as MRTF-A.

The interaction of PALLD with FHOD1 is consistent with its location in the ICD, where it also binds to SORBS2 (*Rönty et al., 2005*). FHOD1 has been reported to be localized at the ICD and costamere of the heart (*Al Haj et al., 2015*; *Dwyer et al., 2014*) consistent with our findings. In contrast, Sanematsu et al., showed that FHOD1 is dispensable for normal cardiac development and function in mouse and failed to detect FHOD1 in the heart (*Sanematsu et al., 2019*). The reason for this discrepancy is unclear as several different FHOD1 antibodies were tested in each of the studies. Nevertheless, the low expression level of FHOD1 in the heart and absence of a pathological cardiac phenotype of FHOD1-deficient mice questions the relevance of the PALLD-FHOD1 interaction in CMCs. In contrast, FHOD1 is highly expressed in SMCs and other cell types, where also PALLD is present, so the interaction is likely to be important in other tissues. FHOD1 suppresses actin polymerization by inhibiting nucleation and elongation but stabilizes actin filaments by capping the actin barbed ends, protecting them from depolymerization, while simultaneously mediating F-actin bundling by connecting them to adjacent actin filaments (*Schönichen et al., 2013*). In line with this, FHOD1 silencing was shown to reduce migration and invasion of breast cancer cells by inhibiting the nuclear translocation of the SRF coactivator MRTF-A, which is sequestered in the cytoplasm by G-actin and regulated by changes in actin dynamics (*Jurmeister et al., 2012*). PALLD directly binds to MRTF-A (*Filomena et al., 2020*; *Jin et al., 2010*) and stabilizes the actin cytoskeleton by promoting actin polymerization and preventing actin depolymerization, consequently promoting MRTF-mediated activation of SRF signaling (*Dixon et al., 2008*; *Filomena et al., 2020*; *Gurung et al., 2016*). Thus, it is possible that PALLD and FHOD1 may cooperate in regulating actin dynamics and SRF signaling.

Global knockout of PALLD in mouse was reported to result in embryonic lethality before E15.5 due to multiple defects, including failure of body walls to close, resulting in exencephaly and herniation of abdominal organs (*Luo et al., 2005*). Furthermore, analysis of PALLD knockout embryos revealed a key role of PALLD in the induction of SMC differentiation in the developing vasculature (*Jin et al.,*

*2010*) and fibroblasts derived from knockout embryos showed defects in stress fiber formation, cell adhesion, and motility (*Luo et al., 2005*). Based on its similarity to MYPN, associated with different forms of cardiomyopathy (*Bagnall et al., 2010*; *Duboscq-Bidot et al., 2008*; *Meyer et al., 2013*; *Purevjav et al., 2012*), as well as high expression levels of specific PALLD isoforms in cardiac muscle both during development and at adult stage, we hypothesized that PALLD plays an important role also in the heart (*Wang and Moser, 2008*).

To determine the role of PALLD in the heart in vivo, we floxed *Palld* exon 14 (see *Figure 1—figure supplement 2*), allowing us to generate conditional (cPKO) and inducible (cPKOi) CMC-specific knockout mice for all PALLD isoforms except for two predicted isoforms not containing exon 14. Due to extensive differential splicing of PALLD, it was not possible to target all isoforms. Unexpectedly, cPKO mice exhibited no pathological cardiac phenotype at basal levels and showed a normal hypertrophic response to TAC. In contrast, cPKOi mice induced at adult stage developed cardiac dilation and systolic dysfunction within 8 wk after TAM induction, associated with interstitial fibrosis and upregulation of cardiac stress markers. Consistently, reduced sarcomere shortening and $Ca^{2+}$ transient amplitude were observed in CMCs, and decreased maximum $Ca^{2+}$-activated isometric tension was found in cPKOi myofibrils, while the kinetics of contraction were unaffected. As biomechanical properties of myofibril preparations from the LV were measured at $Ca^{2+}$ concentrations that cannot be reached under the physiological conditions of cardiac contraction, we cannot exclude that our observations might not fully recapitulate in vivo mechanisms. The heart weight to body weight ratio was unaltered, although CMC size was increased, mainly due to increased CMC width.

At the ultrastructural level, ICDs in cPKOi mice were disorganized and more convoluted compared to those of control mice. The ICD is a highly complex structure joining together adjacent CMCs in the heart and is made up of three major complexes: adherens junctions, where actin filaments are anchored and connect the sarcomere to the cell membrane; desmosomes, which tether the cell membrane to the intermediate filament network; and gap junctions, which permit the passage of ions and small molecules between neighboring CMCs, allowing for electrical and metabolic coupling between cells (reviewed in *Vermij et al., 2017*). In contrast to previous belief, rather than functioning independently, the three junctions cooperate via crosstalk between their components in mixed-type junctions termed '*area composita*' (*Borrmann et al., 2006*; *Franke et al., 2006*). Thus, the ICD can be considered as a single functional unit important for mechanical coupling and transmission of contractile force and electrical signals between adjacent CMCs. PALLD plays a well-known role in the organization of the actin cytoskeleton (*Jin, 2011*; *Otey et al., 2009*) and binds directly to F-actin (*Dixon et al., 2008*), ALP (*von Nandelstadh et al., 2009*), and SORBS2 (*Rönty et al., 2005*), which are associated with the adherens junction (*Ding et al., 2020*; *Pashmforoush et al., 2001*). Thus, destabilization of the adherens junction in cPKOi mice may be responsible for the structural alterations at the ICD, also affecting nearby sarcomeres, as evident by Z-line abnormalities and sarcomere disruption in the vicinity of the adherens junction in high-amplitude ICDs. PALLD also interacts with several components of the transitional junction, where the contractile apparatus is connected the ICD to allow force transmission between adjacent CMCs (*Bennett, 2012*). In particular, titin (*Filomena et al., 2021*), α-actinin (*Bang et al., 2001*; *Rönty et al., 2004*), and cypher (*von Nandelstadh et al., 2009*), all interaction partners of PALLD, are associated with the transitional junction, suggesting a possible effect of PALLD on the attachment of the myofilament to the ICD and the stability of the junction. Immunofluorescence stainings and western blot analyses did not show any changes in the localization of ICD proteins, including the PALLD interaction partners FHOD1 and SORBS2. However, the protein level of SORBS2 was ~1.6-fold upregulated in cPKOi heart. SORBS2, also known as Arg kinase-binding protein 2 (ArgBP2), is a scaffold protein localized at the adherens junction of the ICD as well as the Z-line at lower levels (*Ding et al., 2020*). SORBS2 overexpression in mouse heart was reported to lead to upregulation of β-tubulin, a direct interaction partner of SORBS2, and consequent microtubule densification, resulting in junctophilin 2 translocation, T-tubule disorganization, defective excitation-contraction coupling, and systolic dysfunction (*Li et al., 2020*). Furthermore, SORBS was found to be upregulated in patients with left ventricular non-compaction (LVNC), but not HCM and arrhythmogenic cardiomyopathy, and may play a role in the progression of heart failure in LVNC patients (*Li et al., 2020*). Thus, increased SORBS2 expression in cPKOi mice may contribute to ICD instability and cardiac dysfunction. ICD alterations, such as altered expression of ICD proteins and increased, more variable ICD amplitude, are commonly associated with DCM both in human and mouse (*Wilson et al., 2014*). In addition,

mutations in several genes (*VCL*,*DSP*,*DSG2*, *DES*) encoding ICD proteins have been associated with DCM and several knockout mouse models of adherens junction and desmosomal proteins develop DCM (*Bang et al., 2022*; *Hirschy et al., 2010*; *Kostetskii et al., 2005*; *Li et al., 2012*; *Norgett et al., 2000*; *Sheikh et al., 2006*; *Zemljic-Harpf et al., 2007*). Thus, it is plausible to suggest that structural alteration of the ICD in the absence of PALLD is the primary event leading to DCM and systolic dysfunction in cPKOi mice. Our data has shown that, in addition to the ICD, PALLD localizes to other cellular compartments, including the nucleus. Therefore, we cannot rule out the possibility that ultrastructural changes observed in cPKOi cardiomyocytes might be a consequence of LV dilation rather than the primary cause of the observed phenotype. Interestingly, MKO mice, which developed a mild form of DCM under basal conditions, also showed an increase in ICD fold amplitude, although only by ~27% and without other obvious ICD abnormalities (*Filomena et al., 2021*), suggesting that PALLD plays a more important structural role in the ICD than MYPN.

The absence of a pathological cardiac phenotype in cPKO mice is surprising based on the rather severe cardiac phenotype of cPKOi mice induced at adult stage. Due to the structural homology of PALLD to MYPN, we hypothesized that MYPN can compensate for the absence of PALLD. However, the phenotype of cPKO/MKO dKO mice was indistinguishable from that of MKO mice, strongly suggesting that MYPN does not compensate for the loss of PALLD in cPKO mice. Another possibility is that the small PALLD isoforms not targeted by our knockout approach can compensate for the absence of the other PALLD isoforms when absent during development. However, these isoforms were not upregulated in cPKO mice and do not contain binding sites for most of the PALLD interaction partners, so we consider this rather unlikely. We hypothesize that when PALLD is knocked out during embryonic development, the mice are able to adapt through activation of compensatory mechanisms, whereas this is not anymore possible when PALLD is knocked out in adult mice. This phenomenon has also been observed in other genetic models (reviewed in *El-Brolosy and Stainier, 2017*).

Analysis of *MYPN* and *PALLD* mRNA levels in LV biopsies from human ICM and DCM patients showed an ~2.9-fold upregulation of *MYPN* and an ~1.9-fold upregulation of the structurally similar 200 kDa *PALLD* isoform in DCM hearts, whereas the overall transcript level of all PALLD isoforms except for potential truncated N-terminal transcript(s) was ~2.8- and ~3.0-fold reduced in DCM and ICM patients, respectively. Similarly, in two RNA-Seq studies, *PALLD* was reported to be downregulated in both DCM/non-ischemic cardiomyopathy (NICM) (~1.7- to ~2.0-fold) and ICM (~1.6- to ~2.1-fold) patients, while *MYPN* was upregulated in both DCM/NICM (~1.5- to ~1.6-fold) and ICM (~1.6- to ~1.7-fold) patients (*Sweet et al., 2018*; *Yang et al., 2014*). The absent modulation of *MYPN* in ICM patients in this study may be due to different patient cohorts and detection methods used. The 90 kDa PALLD isoform is the predominant isoform in the heart, while the 200 kDa PALLD isoform is expressed at much lower levels, explaining the upregulation of the 200 kDa *PALLD* isoform despite the overall reduced transcript level of *PALLD*. Notwithstanding the transcriptional modulation of both *MYPN* and *PALLD* in DCM and ICM patients, no changes in protein expression were found, so the altered transcript levels are unlikely to have any functional effects. As previously reported, CARP/Ankrd1 protein levels were significantly increased in DCM patients (*Kempton et al., 2018*; *Nagueh et al., 2004*; *Zolk et al., 2002*).

Altogether, our data demonstrate an important role of PALLD for normal heart function, suggesting PALLD as a possible candidate gene for cardiomyopathy. Based on the absence of a phenotype in cPKO mice, loss-of-function mutations are unlikely to affect cardiac function and would likely lead to embryonic lethality due to the essential role of PALLD in other tissues and cell types. However, as for MYPN, whose ablation does not severely affect cardiac function, it is possible that mutations with dominant negative effects may be linked to cardiac disease. The reason why a putative link between *PALLD* mutations and cardiac disease has not been established could be due to lack of focus on the *PALLD* gene.

## Materials and methods
### Animal experiments
All animal experiments were approved by the Italian Ministry of Health (approval 12/2011) and performed in full compliance with the rules and regulations of the European Union (Directive 2010/63/EU of the European Parliament) and Italy (Council of 22 September 2010; directive from the Italian

Ministry of Health) on the protection of animal use for scientific purposes. Mice used for experiments were sacrificed by cervical dislocation under isoflurane anesthesia. Animals were randomly assigned to different experimental groups before the start of experiments. The investigators were blinded to genotype and treatment.

## Generation of conditional (cPKO) and inducible (cPKOi) cardiac muscle-specific PALLD KO mice

*Palld* genomic DNA was obtained from a 129SVJ mouse genomic library (Stratagene, La Jolla, CA) and used for the generation of a targeting construct containing *loxP* sites flanking exon 14 (**Figure 1—figure supplement 2**) (ggggttcccaaagaagtccagtagaactgcTAGAATTGCCTCTGATGAGGAGATTCAAGGCACAAAGGATGCTGTCATCCAAGACCTGGAACGGAAGCTTCGCTTCAAGGAGGACCTTCTGAACAATGGCCAACCG; NCBI Gene ID: 72333) as well as a neomycin (neo) cassette flanked by Flpase Recognition Target (FRT) sites. Exon 14 was floxed since it is present in the most common PALLD isoforms (90 kDa, 140 kDa, 200 kDa) (see **Figure 1—figure supplement 2**) and contains a number of nucleotides not evenly divisible by 3, so that Cre-mediated recombination would result in a premature stop codon and consequent mRNA degradation by nonsense-mediated decay. The construct was generated in the pBluescript II KS+ vector, and the 5′ arm of homology consisted of a 3983 bp *NotI–EcoRV* fragment (5′ forward: gctccaccgcggtggcggccgc/AGAGCAGTTATCCTAAG; 5′ reverse: gttatatt aagggttccggatcgatgatatc/AACATGAAATG) fused with a *loxP* site upstream of *Palld* exon 14. The 3′ arm of homology was a 3062 kb *SalI–SalI* fragment (3′ forward: ccaagctgatcctctagagtcgac/TCCATGAGGCTCTGTC, and 3′ reverse: cgggcccccccctcgaggtcgac/GGAAAGGAAAACACAG) located downstream of exon 14 followed by a FRT-neo-FRT cassette and a second *loxP* site (**Figure 2—figure supplement 1**). The Diphteria toxin A fragment (DTA) gene was inserted for negative selection. The final targeting construct was verified by sequencing and linearized with *NotI* before electroporation into R1 embryonic stem (ES) cells at the Transgenic Core Facility at the University of California San Diego. G418-resistant ES clones were screened for homologous recombination by Southern blot analysis of *EcoRV*-digested ES cell DNA with a 471 bp probe generated by PCR on mouse genomic DNA with *palld*-specific primers (sense: ATTCTTGAATGTATGGTGCCCTTGA; reverse: CTCAAAGCAGACCTCATCACAAAAC) (see **Figure 2—figure supplement 1A and B**). The wild-type allele is represented by the band of 19,607 bp, whereas the 10,130 bp band represents the correctly targeted mutant allele. One clone out of 480 G418-resistant ES clones that had undergone homologous recombination was identified. The homologous recombinant ES clone was microinjected into C57BL/6J blastocysts and transferred into pseudopregnant mice. Male chimeras were bred with female C57BL/6J mice to generate germ line transmitted heterozygous *Palld* floxed (*Palld*$^{fl/+}$) mice, which were crossed with FLPe deleter mice (**Kanki et al., 2006**) to remove the *neo* gene and subsequently backcrossed for six generations with C57BL/6J mice. To generate conditional cardiac-specific knockout (cPKO) mice, homozygous *Palld* floxed (*Palld*$^{fl/fl}$) mice were interbreed with *Myh6*-nuclear-Cre (**Abel et al., 1999**) mice. The resulting *Palld*$^{fl/+}$;*Myh6*$^{Cre/+}$ mice were crossed with *Palld*$^{fl/+}$ mice to generate *Palld*$^{fl/fl}$;*Myh6*$^{Cre/+}$ mice as cPKO mice and *Palld*$^{fl/fl}$;*Myh6*$^{+/+}$ (*Palld*$^{fl/fl}$) mice as littermate controls. Furthermore, *Palld*$^{+/+}$;*Myh6*$^{Cre/+}$ (*Myh6*$^{Cre/+}$) mice derived from the same crossings were used as additional controls. Genotyping was performed on mouse tail DNA using PALLD specific primers (sense: GCTTCGCTTCAAGGAGGACCTTCTG; reverse: TGTATATCATGTTGTGGTGTCAGCC), giving rise to a 348 band for the wild-type allele and a 418 band for the targeted allele (**Figure 2—figure supplement 1C**). The presence of Cre was verified using Cre-specific primers (sense: ACGTTCACCGGCATCAACGT, reverse: CTGCATTACCGGTCGATGCA), giving rise to a 356 bp band (**Figure 2—figure supplement 1C**). Inducible cardiac-specific knockout (cPKOi) mice were generated by interbreeding of *Palld*$^{fl/fl}$ mice with *Myh6*-MerCreMer (MCM) transgenic mice (**Sohal et al., 2001**). Knockout of *Palld* in cPKOi (*Palld*$^{fl/fl}$;*Myh6*$^{MCM/+}$) male mice was induced at 7 wk of age by intraperitoneal injection of TAM dissolved in sesame oil (30 mg/kg/day) for 5 days, a dose that does not produce toxicity in mice (**Rouhi et al., 2022**). TAM-treated *Palld*$^{fl/fl}$;*Myh6*$^{+/+}$ (*Palld*$^{fl/fl}$) and *Palld*$^{+/+}$;*Myh6*$^{MCM/+}$ (*Myh6*$^{MCM/+}$) mice as well as *Palld*$^{fl/fl}$;*Myh6*$^{MCM/+}$ mice not receiving TAM served as controls. Double knockout mice (cPKO/MKO dKO) for PALLD and MYPN were generated by interbreeding of *Palld*$^{fl/fl}$;*Myh6*$^{Cre/+}$ mice with *Mypn*$^{-/-}$ mice, giving rise to *Mypn*$^{-/-}$/*Palld*$^{fl/fl}$;*Myh6*$^{Cre/+}$ mice as cPKO/MKO dKO mice and *Mypn*$^{-/-}$/*Palld*$^{fl/fl}$ mice control mice derived from the same crosses. All experiments were performed on male mice as females often develop a less severe cardiac phenotype due to the cardioprotective role of estrogen (**Brower et al., 2003**; **Du, 2004**).

## In vivo cardiac physiology

Mice anaesthetized by inhalation of 1% isoflurane were subjected to transthoracic echocardiography using a Vevo 2100 System (VisualSonics) and a 30 MHz probe as previously described (*Tanaka et al., 1996*). Transverse Aortic Constriction (TAC) was executed with a 27-gauge needle on 2-mo-old cPKO and control mice anesthetized by intraperitoneal injection of a mixture of ketamine (100 mg/kg) and xylazine (5 mg/kg) as described (*Tanaka et al., 1996*). Cardiac morphology and function were assessed by transthoracic echocardiography and the pressure gradient was evaluated by Doppler echocardiography. Only mice with a pressure gradient >70 mmHg were included in the analysis. Sham-operated mice were used as controls.

## Human tissue

Human myocardial tissue was collected from the apex of patients with ICM or DCM during left ventricular assist device (LVAD) implantation at Leipzig Heart Center in Germany following approval by the institutional review board (protocol #240/16-ek) and signed informed consent from the patients according to the principles of the Declaration of Helsinki. Myocardial tissue from healthy donors rejected for transplantation was obtained from Careggi University Hospital, Florence, Italy (protocol #2006/0024713; renewed May 2009). All patients were Caucasian. Myocardial tissue was immediately snap frozen in liquid nitrogen and stored at –80°C for further processing.

## DNA constructs

Human *PALLD*, *MYPN*, *FHOD1*, and *ANKRD1* cDNAs, isolated by PCR using available constructs or a human cDNA library as a template, were cloned into pGBKT7 DNA-BD (Takara Bio), pGADT7-AD (Takara Bio), pFN21A HaloTag CMV Flexi (Promega), pNLF1-N [CMV Hygro] (Promega), and pNLF1-C [CMV Hygro] (Promega) vectors using restriction cloning or the In-Fusion HD Cloning Plus kit (Takara Bio) according to the manufacturer's instructions. Primer sequences are listed in *Supplementary file 1*. All constructs were confirmed by sequencing.

## Yeast two-hybrid assays

Yeast two-hybrid screening was performed using the Matchmaker Gold Yeast Two-Hybrid System (Takara Bio) as described by the manufacturer (*Young et al., 1998*). Briefly, cDNA encoding the N-terminal region of human PALLD comprising Ig1 and Ig2 (amino acids 1–528) was cloned into the pGBKT7 DNA-BD bait vector (see *Supplementary file 1*) and transformed into the yeast two-hybrid Gold yeast strain. The bait stain was combined with a Mate & Plate Human Heart library (Takara Bio) for 24 hr after which cells were plated on selective synthetic defined (SD) agar plates lacking tryptophan, leucine, histidine, and adenine and containing 120 ng/mL Aureobasidin A (SD/–Ade/–His/–Leu/–Trp/AbA). Colonies appearing after 3–4 d of incubation at 30°C were restreaked onto SD/–Ade/–His/–Leu/–Trp/AbA/X-α-Gal plates containing 40 µg/mL X-α-Gal, after which library plasmids from blue clones were isolated and sequenced. To confirm the interactions and narrow down the binding sites, pGBKT DNA-BD and pGADT7-AD vectors (Takara Bio) containing cDNAs encoding regions of MYPN, PALLD, CARP, and FHOD1 were cotransformed into the yeast two-hybrid Gold yeast strain and spotted on SD/–Ade/–His/–Leu/–Trp/AbA/X-α-Gal plates. Interaction was verified after 3–4 d of incubation at 30°C. Successful transformation of the two plasmids was confirmed by growth on SD/-Trp/-Leu plates. Possible autoactivation of bait and prey constructs was tested by cotransformation of the bait or prey vector with empty prey or bait vector, respectively.

## Cell line

Human embryonic kidney 293 (HEK293) cells were maintained in Dulbecco's modified Eagle's medium (DMEM) supplemented with 4% fetal bovine serum (FBS), 1% L-glutamine, and 1% penicillin/streptomycin. Cells were cultured in a 37°C incubator with 5% $CO_2$ and routinely for mycoplasma contamination.

## NanoBRET protein:protein interaction assay

HEK293 cells cotransfected with pFN21A HaloTag CMV Flexi vector (Promega) expressing PALLD and pNLF1-N [CMV Hygro] or pNLF1-C [CMV Hygro] vector (Promaga) expressing ANKRD1 or FHOD1 were treated with 100 nM HaloTag NanoBRET 618 Ligand (Promega), whereafter signals were

detected 6 hr later using a Synergy 4 instrument (BioTek). Results were analyzed using GraphPad Prism 9.0 software.

## RNA extraction and quantitative qRT-PCR

LV RNA from mice or patients was extracted using PureZOL RNA isolation reagent (Bio-Rad Laboratories) according to the manufacturer's instructions. For qRT-PCR, first-strand cDNA synthesis was performed using the High Capacity cDNA Reverse Transcription kit (Thermo Fisher Scientific), whereafter qRT-PCR was performed in triplicate with custom designed oligos (see *Supplementary file 1*) using the SYBR Select Master Mix (Thermo Fisher Scientific). Relative expression analysis was performed using the ΔΔCt method using *Gapdh* (mice) or *PGK1* (human) for normalization.

## Isolation of adult ventricular CMCs

For isolation of ventricular CMCs, hearts were cannulated and mounted on a Langendorff perfusion apparatus and perfused with Hank's Balanced Salt Solution (HBSS w/o $CaCl_2$ and $MgCl_2$) supplemented with 1.2 mM $MgSO_4$, 15 mM glucose, 30 mM Taurine, and 1 mM $MgCl_2$ hexahydrate for 10 min at 37°C. Liberase blendzyme (Roche) (50 µg/ml) was then added to the solution and perfusion was continued for about 10 min until the heart became flaccid. Subsequently, the heart was removed and cells were dissociated and filtered through a 70 µm filter after which bovine serum albumin (Merck Life Science) was added to a final concentration of 4% to inactivate the enzyme. The cells were then allowed to settle and resuspended in fresh solution. For determination of CMC size, length, and width, pictures were taken on an Olympus BX53 Fluorescent microscope and analyzed using ImageJ, version 2.1.0/1.53C (NIH).

## CMC contractility and intracellular $Ca^{2+}$ transient measurements

Simultaneous measurements of CMC contractility and $Ca^{2+}$ transients were carried out on an IonOptix system as previously described (*Kondo et al., 2006*). Briefly, CMCs loaded with 1 µM of the $Ca^{2+}$ probe Fura-2 AM (Thermo Fisher Scientific) were placed in a perfusion system and continuously perfused with perfusion buffer (HBSS without $Ca^{2+}$ and $Mg^{2+}$, supplemented with 1.2 mM $MgSO_4$, 15 mM glucose, 30 mM taurine, and 1.0 mM $MgCl_2$), containing 1.0 mM $CaCl_2$ at 37°C at a constant flow rate of 1 mL/min. Loaded cells were paced (25 V) via two electrodes at frequencies of 0.5, 1.0, and 2.0 Hz, and sarcomere shortening and Fura-2 ratio (measured at 512 nm upon excitation at 340 and 380 nm) were simultaneously recorded on a Nikon Eclipse TE-2000S inverted fluorescence microscope with a ×40/1.3 N.A. objective and an attached CCD camera (MyoCam-S, IonOptix). Data acquisition and analysis were performed using Ion Wizard software, version 6.6.11 (IonOptix).

## Mechanical experiments in isolated myofibrils

Mechanical data were collected at 15°C from small bundles of cardiac myofibrils from frozen ventricular strips of cPKOi and control mice as previously described (*Kreutziger et al., 2011*). Briefly, thin myofibril bundles (1–4 µm width, initial sarcomere length around 2.2 µm) were maximally $Ca^{2+}$-activated (pCa 4.5) and fully relaxed (pCa 8.0) by fast solution switching. Maximal tension and the kinetics of force activation, and force relaxation were measured.

## Histology and immunofluorescence stainings

For histology, mouse hearts were harvested, relaxed in 50 mM KCl in PBS, and fixed overnight in 4% paraformaldehyde (PFA) in PBS. Subsequently, hearts were dehydrated, embedded in paraffin, and cut in 8 µm sections in the four-chamber view. Briefly, heart sections were stained with hematoxylin and eosin or Picro Sirius Red and imaged using a VS20 DotSlide Digital Virtual Microscopy System (Olympus). The area of fibrosis in the LV was determined using ad hoc software automatically detecting Picro Sirius Red-stained areas based on RGB color segmentation (*Grizzi et al., 2019*). The sum of Picro Sirius Red-stained areas was expressed as a percentage of the LV area excluding unfilled and tissue-free spaces. TUNEL staining was performed using the FragEL DNA Fragmentation Detection kit, Colorimetric – TdT Enzyme (Merck). For immunostaining of cryosectioned heart, the upper half of the heart was relaxed in 50 mM KCl in PBS, fixed for 15 min in 4% PBS, and subsequently saturated in 15 and 30% sucrose in PBS and frozen in OCT. 10 µm sections were permeabilized and blocked for 1 hr in blocking solution containing 5% normal goat serum, 0.3% Triton X-100, and 50 mM

glycine in PBS after which sections were incubated with primary antibodies in wash buffer (blocking buffer diluted 10 times in PBS) overnight at 4°C. After washing in wash buffer, sections were incubated at room temperature for 4 hr with rhodamine-labeled phalloidin (1:100, Merck Life Science) and/or Alexa-Fluor-488 or -568-conjugated IgG secondary antibodies (1:500, Thermo Fisher Scientific). The primary and secondary antibodies used are listed in the Key Resources Table. For immunostaining of adult CMCs, CMCs were fixed for 5 min in 4% PFA in PBS and subsequently plated on laminin-coated 8-well Nunc Lab-Tek Chamber Slides (Thermo Fisher Scientific). CMCs were permeabilized and blocked for 1 hr in blocking solution containing 5% normal goat serum and 0.2% Triton X-100 in PBS after which sections were incubated with primary antibodies (see Key Resources Table) in wash buffer (blocking buffer diluted 10 times in PBS) overnight at 4°C. After washing in PBS, CMCs were incubated at room temperature for 4 hr with rhodamine-labeled phalloidin (1:100, Merck Life Science) and/or Alexa-Fluor-488, -568, or -647-conjugated IgG secondary antibodies (1:500, Thermo Fisher Scientific). Conventional confocal microscopy was performed on a Leica SP8 inverted confocal microscope with a ×60 oil immersion lens, while STED microscopy was performed on a Leica SP8 STED3X SMD/FCS confocal microscope. Individual images (1024 × 1024) were converted to tiff format and merged as pseudocolor RGB images using ImageJ (NIH).

## Transmission electron microscopy (TEM)

For TEM, hearts from cPKOi and control mice 8 and 24 wk after TAM induction were excised and fixed in 3.5% glutaraldehyde in 0.15 M sodium cacodylate buffer, pH 7.4, as described (*Boncompagni et al., 2009a*; *Boncompagni et al., 2009b*). Small bundles of fibers teased from the papillary muscles were then post-fixed in 2% $OsO_4$ in 0.1 M sodium cacodylate buffer, pH 7.4 for 2 hr and block-stained in saturated uranyl acetate. After dehydration, specimens were embedded in an epoxy resin (Epon 812). Ultrathin sections (~50 nm) were cut using a Leica Ultracut R microtome (Leica Microsystem, Austria) with a Diatome diamond knife (DiATOME) and double-stained with uranyl acetate and lead citrate. Sections were viewed in a FP 505 Morgagni Series 268D electron microscope (FEI Company), equipped with a Megaview III digital camera and Soft Imaging System at 60 kV. The ICD fold amplitude was measured at several locations for each ICD on transmission electron micrographs of longitudinally sectioned papillary muscle at a magnification of ×11.000 using the TEM Analysis Software. The ICD amplitude was defined as the width of the fold region as indicated in *Figure 5B*. Sarcomere and I-band lengths were determined on each micrograph in the proximity of the ICD to exclude micrographs presenting shrinkage of the specimens due to fixation and/or longitudinal distortion caused during the cutting of the section.

## SDS-PAGE and western blot analysis

For western blot analysis, LV tissue or adult CMCs were homogenized in RIPA buffer containing 50 mM Tris pH 7.5, 150 mM NaCl, 0.5 mM DTT, 1 mM EDTA, 1% (v/v) SDS, 1% (v/v) Triton X-100, 1 mM PMSF, Roche Complete Protease Inhibitor Cocktail (Thermo Fisher Scientific), and Pierce Phosphatase Inhibitor Mini Tablets (Thermo Fisher Scientific) using a TissueLyser II (QIAGEN). Protein concentration was determined using the Bio-Rad DC Protein Assay Kit (Bio-Rad Laboratories) according to the manufacturer's instructions. SDS-PAGE was performed using standard gels or TGX Stain-Free gels (Bio-Rad Laboratories) after which proteins were transferred to PVDF membranes. Stainings were performed using the primary and secondary antibodies listed in the Key Resources Table. The Immobilon Western Chemiluminescent HRP Substrate (Merck Life Science) was used and chemiluminescence was detected on a Chemidoc MP System (Bio-Rad Laboratories). Relative protein expression was determined by densitometry using Image Lab, version 6.1.0 software (Bio-Rad Laboratories). Normalization was performed to GAPDH or total protein content as determined on UV-activated TGX Stain-Free gels.

## Titin isoform separation and phospho-titin analysis by western blot analysis

Separation of titin isoforms by SDS-PAGE and western blot analyses were performed as previously described (*Hamdani et al., 2013*). Briefly, LV tissue was solubilized in 50 mM Tris-SDS buffer (pH 6.8) containing 8 μg/mL leupeptin (Merck Life Science) and phosphatase inhibitor cocktail (Merck Life Science). SDS-PAGE was performed on 1.8% polyacrylamide/1% agarose gels run at 5 mA for 16 hr,

whereafter titin bands were visualized by Coomassie blue staining. Titin isoform ratio was calculated as the densitometric value for titin N2BA over N2B. For determination of total phosphoserine/threonine phosphorylation and site-specific titin phosphorylation, Western blots were performed using anti-phosphoserine/threonine antibody (ECM Biosciences #PP2551; 1:500) as well as titin phosphosite-specific antibodies against pTTN-Ser3991 (N2B region; phosphorylated by PKA and ERK2; 1:500), pTTN-Ser4080 (N2B region; phosphorylated by PKG; 1:500), and pTTN-Ser12742 (PEVK region; phosphorylated by PKCα; 1:500) (*Kötter et al., 2013*). Densitometry was performed by normalization to total protein content as determined by Coomassie blue staining of each blot.

## Statistical analysis

Sample sizes with adequate power to detect statistical differences between groups were determined based on our previous experience and gold standards in the field. The only exclusion criteria were technical failure or death/injury. Data are represented as mean ± standard error of the mean (SEM). Statistical comparisons between two groups were done using the unpaired Student's *t*-test. Comparisons between multiple groups were performed by one-way or two-way ANOVA with Šídák's or Tukey's multiple comparisons test, as indicated. The Shapiro–Wilk test was performed to confirm normal distribution in each group and in residuals from a linear regression model, Bartlett's test to check for homogeneity of variance across groups, and Spearman's rank correlation test to confirm heteroscedasticity of residuals. The residuals diagnostic was performed with the DHARMa package, version 0.4.1 in R (*R Development Core Team, 2021*). When necessary, data were transformed to meet ANOVA assumptions. For the statistical analyses of echocardiographic parameters over time, a linear mixed model with Tukey's multiple comparisons test was used. The statistical analysis of data not showing equal standard deviation was performed by Brown–Forsythe and Welch's ANOVA with Dunnett's T3 multiple comparisons test (n < 50 per group; qRT-PCR data from human biopsies) or Games–Howell's multiple comparisons test (n > 50 per group; ICD fold amplitude). The functional comparisons of sarcomere shortening and $Ca^{2+}$ transients in CMCs were performed using two-level hierarchical testing with Bonferroni correction to eliminate the effects of variations both within cells and between mice (*Sikkel et al., 2017*). $p < 0.05$ was considered statistically significant. Statistical analysis was performed using Prism 9 (GraphPad) or RStudio, version 1.2.5019 (*RStudio Team, 2020*) software.

## Acknowledgements

We thank Dr. Ju Chen (University of California San Diego, La Jolla, CA), who provided advice and support for the generation of *Palld*[fl/fl] mice, and Marion von Frieling-Salewsky (University of Münster, Münster, Germany) for performing titin gel electrophoresis and Western blot analysis. This work was supported by the Italian Telethon foundation (GGP12282 to MLB); the Italian Ministry of Education, Universities and Research (MiUR PRIN 2010–2011; 2010R8JK2X_006 to MLB); and the European Union's Horizon 2020 research and innovation program (SILICOFCM; 777204 to CP).

## Additional information

### Funding

| Funder | Grant reference number | Author |
| --- | --- | --- |
| Fondazione Telethon | GGP12282 | Marie-Louise Bang |
| Ministero dell'Università e della Ricerca | 2010R8JK2X_006 | Marie-Louise Bang |
| Horizon 2020 Framework Programme | 777204 | Corrado Poggesi |

The funders had no role in study design, data collection and interpretation, or the decision to submit the work for publication.

## Author contributions
Giuseppina Mastrototaro, Formal analysis, Investigation, Writing – original draft; Pierluigi Carullo, Jianlin Zhang, Beatrice Scellini, Nicoletta Piroddi, Simona Nemska, Chiara Tesi, Formal analysis, Investigation; Maria Carmela Filomena, Investigation; Simone Serio, Formal analysis; Carol A Otey, Conceptualization, Resources; Fabian Emrich, Resources, Provided patient biopsies; Wolfgang A Linke, Resources, Supervision; Corrado Poggesi, Resources, Supervision, Funding acquisition, Writing – original draft; Simona Boncompagni, Formal analysis, Investigation, Writing – original draft, Writing - review and editing; Marie-Louise Bang, Conceptualization, Resources, Formal analysis, Supervision, Funding acquisition, Investigation, Methodology, Writing – original draft, Project administration, Writing - review and editing

## Author ORCIDs
Simone Serio ⓘ http://orcid.org/0000-0002-7294-2094
Wolfgang A Linke ⓘ http://orcid.org/0000-0003-0801-3773
Simona Boncompagni ⓘ http://orcid.org/0000-0001-5308-5069
Marie-Louise Bang ⓘ http://orcid.org/0000-0001-8859-5034

## Ethics
Human myocardial biopsies from cardiomyopathy patients were obtained from Leipzig Heart Center, Germany following approval by the institutional review board (protocol #240/16-ek) and signed informed consent from the patients according to the principles of the Declaration of Helsinki. Myocardial biopsies from healthy donors rejected for transplantation were obtained from Careggi University Hospital, Florence, Italy (protocol #2006/0024713; renewed May 2009).

All animal studies were approved by the Italian Ministry of Health and performed in full compliance with the rules and regulations of the European Union (Directive 2010/63/EU of the European Parliament) and Italy (Council of 22 September 2010; directive from the Italian Ministry of Health) on the protection of animals used for scientific purposes.

## Decision letter and Author response
Decision letter https://doi.org/10.7554/eLife.78629.sa1
Author response https://doi.org/10.7554/eLife.78629.sa2

---

# Additional files

## Supplementary files
• Supplementary file 1. Oligos used for quantitative real-time PCR (qRT-PCR) and clonings.
• MDAR checklist

## Data availability
All data generated and analysed during this study are included in the manuscript and figure supplements. Source data files have been provided for all figures.

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

## Appendix 1

**Appendix 1—key resources table**

| Reagent type (species) or resource | Designation | Source or reference | Identifiers | Additional information |
|---|---|---|---|---|
| Strain, strain background (male *Mus musculus*) | MYPN knockout (MKO) mice (C57BL/6J background) | *Filomena et al., 2020* | N/A | |
| Strain, strain background (male *M. musculus*) | PALLD floxed mice (C57BL/6J background) | This paper | N/A | |
| Strain, strain background (male *M. musculus*) | *Myh6*-nuclear-Cre mice (C57BL/6J background) | *Abel et al., 1999* | N/A | |
| Strain, strain background (male *M. musculus*) | *Myh6*-MerCreMer transgenic mice (C57BL/6J background) | The Jackson Laboratories | Cat# 005657 RRID:IMSR_JAX:005657 | |
| Strain, strain background (*M. musculus*) | C57BL/6J | The Jackson Laboratories | Cat# 000664; RRID:IMSR_JAX:000664 | |
| Strain, strain background (*Escherichia coli*) | DH5α electrocompetent cells | New England BioLabs | Cat# C2989K | |
| Strain, strain background (*Saccharomyces cerevisiae*) | Y2H Gold yeast strain | Takara Bio | Cat# 630498 | |
| Strain, strain background (*S. cerevisiae*) | L40 yeast strain | Takara Bio | N/A | |
| Strain, strain background (*Homo sapiens*) | HEK293 | ATCC | Cat# CRL-1573 | |
| Antibody | Anti-PALLD 622 (rabbit polyclonal) | *Pogue-Geile et al., 2006*; *Goicoechea et al., 2010* | Kindly provided by Prof. Carol Otey, University of North Carolina, Chapel Hill, NC, USA | WB (1:500) IF (1:30) |
| Antibody | Anti-PALLD Proteintech (rabbit polyclonal) | Proteintech Group | Cat# 10853-1-AP | WB (1:1000) IF (1:30) |
| Antibody | Anti-PALLD 1E6 (mouse monoclonal) | *Parast and Otey, 2000* Novus Biochemicals | Cat# NBP1-25959 | WB (1:1000) IF (1:20) |
| Antibody | Anti-PALLD 4D10 (mouse monoclonal) | *Parast and Otey, 2000* | N/A | IF (1:10) |
| Antibody | Anti-MYPN (rabbit polyclonal) | *Yamamoto et al., 2013* | N/A | WB (1:1000) |
| Antibody | Anti-nebulette (rabbit polyclonal) | *Mastrototaro et al., 2015* | N/A | WB (1:200) IF (1:20) |
| Antibody | Anti-α-actinin (mouse monoclonal) | Merck Life Science | Cat# A7811; RRID:AB_476766 | WB (1:50000) IF (1:250) |
| Antibody | Anti-ANKRD1/CARP (rabbit polyclonal) | *Miller et al., 2003* | N/A | WB (1:200) IF (1:20) |

*Appendix 1 Continued on next page*

*Appendix 1 Continued*

| Reagent type (species) or resource | Designation | Source or reference | Identifiers | Additional information |
|---|---|---|---|---|
| Antibody | Anti-Cypher (rabbit polyclonal) | *Zhou et al., 1999* | Kindly provided by Prof. Ju Chen, University of California San Diego, La Jolla, CA, USA | WB (1:500) IF (1:20) |
| Antibody | Anti-MKL1/MRTF-A (rabbit polyclonal) | Santa Cruz Biotechnology | Cat# sc-21558 | WB (1:500) IF (1:20) |
| Antibody | Anti-MKL1/MRTF-A (rabbit polyclonal) | Immunological Sciences | Cat# AB-84312; RRID:AB_2892156 | WB (1:500) |
| Antibody | Calpain 3 (rabbit polyclonal) | *König et al., 2003* | Kindly provided by Prof. Ahmed Ouali, QuaPA, INRA de Clermont Ferrand-Theix, St Genès Champanelle, France | WB (1:500) |
| Antibody | Anti-desmin (rabbit polyclonal) | Abcam | Cat# Ab8592; RRID:AB_306653 | IF (1:80) |
| Antibody | Anti-SORBS2 (mouse monoclonal) | Merck Life Science | Cat# SAB4200183; RRID:AB_10638778 | WB (1:750) IF (1:30) |
| Antibody | Anti-FHOD1 (goat polyclonal) | Santa Cruz Biotechnology | Cat# sc-46965; RRID:AB_2247011 | WB (1:500) IF (1:30) |
| Antibody | Anti-desmoplakin 1/2 (mouse monoclonal) | Bio-Rad Laboratories | Cat# 2722-5204; RRID:AB_619950 | WB (1:750) IF (1:20) |
| Antibody | Anti-N-cadherin (mouse monoclonal) | Cell Signaling Technology | Cat# 4061; RRID:AB_10694647 | WB (1:1000) IF (1:80) |
| Antibody | Anti-α-E-catenin (mouse monoclonal) | Santa Cruz Biotechnology | Cat# sc-9988; RRID:AB_626805 | WB (1:1000) IF (1:20) |
| Antibody | Anti-β-catenin (rabbit monoclonal) | Cell Signaling Technology | Cat# 8480; RRID:AB_11127855 | WB (1:1000) IF (1:25) |
| Antibody | Anti-γ-catenin/ plakoglobin (rabbit polyclonal) | Immunological Sciences | AB-90215; RRID:AB_2892157 | WB (1:1000) IF (1:20) |
| Antibody | Anti-vinculin (mouse monoclonal) | Merck Life Science | Cat# V9264; RRID:AB_10603627 | WB (1:2000) IF (1:80) |
| Antibody | Anti-desmocollin 2 (rabbit monoclonal) | Fitzgerald | Cat# 20R; RRID: DR004AB_1284102 | IF (1:50) |
| Antibody | Anti-desmoglein 1+2 (mouse monoclonal) | Fitzgerald | Cat# 10R-D105A; RRID:AB_1284107 | WB (1:500) IF (1:30) |
| Antibody | Anti-plakophilin 2 (mouse monoclonal) | Fitzgerald | Cat# 10R-P130B; RRID: AB_1288393 | IF (1:50) |
| Antibody | Anti-connexin 43 (mouse monoclonal) | Thermo Fisher Scientific | Cat# 35-5000; RRID:AB_87322 | WB (1:400) IF (1:50) |
| Antibody | Anti-Smad1/5-Ser463/465/ Smad8-Ser426/428 (rabbit polyclonal) | Cell Signaling Technology | Cat# 9511; RRID:AB_331671 | WB (1:1000) |
| Antibody | Anti-PKCα (rabbit polyclonal) | Cell Signaling Technology | Cat# 2056; RRID:AB_2284227 | WB (1:1000) |
| Antibody | Anti-pPDK-Ser241 (rabbit polyclonal) | Cell Signaling Technology | Cat# 3438; RRID:AB_2161134 | WB (1:1000) |

*Appendix 1 Continued on next page*

*Appendix 1 Continued*

| Reagent type (species) or resource | Designation | Source or reference | Identifiers | Additional information |
|---|---|---|---|---|
| Antibody | Anti-Akt (rabbit polyclonal) | Cell Signaling Technology | Cat# 9272; RRID:AB_329827 | WB (1:1000) |
| Antibody | Anti-pAkt-Thr308 (rabbit polyclonal) | Cell Signaling Technology | Cat# 2965; RRID:AB_2255933 | WB (1:500) |
| Antibody | Anti-pAkt-Ser473 (rabbit polyclonal) | Cell Signaling Technology | Cat# 4060; RRID:AB_2315049 | WB (1:500) |
| Antibody | Anti-pGSK3β-Ser9 (rabbit polyclonal) | Cell Signaling Technology | Cat# 5558; RRID:AB_10013750 | WB (1:500) |
| Antibody | Anti-GSK3β (rabbit polyclonal) | Cell Signaling Technology | Cat# 9315; RRID:AB_490890 | WB (1:1000) |
| Antibody | Anti-pP70-Ser6K-Thr421/ Ser424 (rabbit polyclonal) | Cell Signaling Technology | Cat# 9204; RRID:AB_2265913 | WB (1:500) |
| Antibody | Anti-P70-Ser6K (rabbit polyclonal) | Cell Signaling Technology | Cat# 2708; RRID:AB_390722 | WB (1:500) |
| Antibody | Anti-pMEK1/2-Ser217/221 (rabbit polyclonal) | Cell Signaling Technology | Cat# 9154; RRID:AB_2138017 | WB (1:500) |
| Antibody | Anti-pErk1/2-Thr202/Tyr204 (rabbit polyclonal) | Cell Signaling Technology | Cat# 4370; RRID:AB_2315112 | WB (1:500) |
| Antibody | Anti-Erk1/2 (mouse monoclonal) | Santa Cruz Biotechnology | Cat# sc-514302; RRID:AB_2571739 | WB (1:1000) |
| Antibody | Anti-pP38-Thr180/Tyr182 (rabbit polyclonal) | Cell Signaling Technology | Cat# 4631; RRID:AB_331765 | WB (1:500) |
| Antibody | Anti-P38α/β (mouse monoclonal) | Santa Cruz Biotechnology | Cat# sc-7972; RRID:AB_628079 | WB (1:500) |
| Antibody | Anti-phosphoserine/ threonine (rabbit polyclonal) | ECM Biosciences | Cat# PP2551; RRID:AB 1184778 | WB (1:500) |
| Antibody | Anti-phosphoserine/ threonine (rabbit polyclonal) | ECM Biosciences | Cat# PP2551; RRID:AB_1184778 | WB (1:500) |
| Antibody | Anti-pTTN-Ser3991 (rabbit polyclonal) | *Kötter et al., 2013* | N/A | WB (1:500) |
| Antibody | Anti-pTTN-Ser4080 (rabbit polyclonal) | *Kötter et al., 2013* | N/A | WB (1:500) |
| Antibody | Anti-pTTN-Ser12742 (rabbit polyclonal) | *Kötter et al., 2013* | N/A | WB (1:500) |
| Antibody | Anti-GAPDH (rabbit polyclonal) | Proteintech | Cat# 10494-1-AP; RRID:AB_2263076 | WB (1:15000) |
| Antibody | Goat anti-mouse IgG (H+L) Highly-cross Adsorbed Secondary antibody, Alexa Fluor 488-conjugated IgG (goat polyclonal) | Thermo Fisher Scientific | Cat# A11029; RRID:AB_138404 | IF (1:500) |
| Antibody | Goat anti-rabbit IgG (H+L) Highly-cross Adsorbed Secondary antibody, Alexa Fluor 488 (goat polyclonal) | Thermo Fisher Scientific | Cat# A11034; RRID:AB_2576217 | IF (1:500) |

*Appendix 1 Continued on next page*

*Appendix 1 Continued*

| Reagent type (species) or resource | Designation | Source or reference | Identifiers | Additional information |
|---|---|---|---|---|
| Antibody | Goat anti-mouse IgG (H+L) Highly-cross Adsorbed Secondary antibody, Alexa Fluor 568 (goat polyclonal) | Thermo Fisher Scientific | Cat# A11031; RRID:AB_144696 | IF (1:500) |
| Antibody | Goat anti-rabbit IgG (H+L) Highly-cross Adsorbed Secondary antibody, Alexa Fluor 568 (goat polyclonal) | Thermo Fisher Scientific | Cat# A11036; RRID:AB_10563566 | IF (1:500) |
| Antibody | Goat anti-mouse IgG (H+L) Highly-cross Adsorbed Secondary antibody, Alexa Fluor 647 (goat polyclonal) | Thermo Fisher Scientific | Cat# A21236; RRID:AB_2535805 | IF (1:500) |
| Antibody | Goat anti-rabbit IgG (H+L) Highly-cross Adsorbed Secondary antibody, Alexa Fluor 647 (goat polyclonal) | Thermo Fisher Scientific | Cat# A21245; RRID:AB_2535813 | IF (1:500) |
| Antibody | Goat anti-rabbit IgG Horseradish Peroxidase (HRP) (goat polyclonal) | Thermo Fisher Scientific | Cat# 31460; RRID:AB_228341 | WB (1:5000) |
| Antibody | Goat anti-mouse IgG-HRP (goat polyclonal) | Thermo Fisher Scientific | Cat# 31430; RRID:AB_228307 | WB (1:5000) |
| Antibody | Donkey anti-goat IgG-HRP (donkey polyclonal) | Santa Cruz Biotechnology | Cat# sc-2020; RRID:AB_631728 | WB (1:2000) |
| Recombinant DNA reagent | pGBKT7 DNA-BD vector | Takara Bio | Cat# 630443 | |
| Recombinant DNA reagent | pGADT7 AD vector | Takara Bio | Cat# 630442 | |
| Recombinant DNA reagent | Mate & Plate Human Heart library | Takara Bio | Cat# 630471 | |
| Recombinant DNA reagent | pGBKT7 human MYPN N-term start-Ig2 (bp 233–1798; aa. 1–522; NM_032578.3) | This paper | N/A | Cloning primers in *Supplementary file 1* |
| Recombinant DNA reagent | pGBKT7 human MYPN C-term Ig3-end (bp 3044–4195; aa. 938–1320; NM_032578.3) | This paper | N/A | Cloning primers in *Supplementary file 1* |
| Recombinant DNA reagent | pGBKT7 human MYPN full-length | This paper | N/A | Cloning primers in *Supplementary file 1* |
| Recombinant DNA reagent | pGBKT7 human PALLD full-length | This paper | N/A | Cloning primers in *Supplementary file 1* |
| Recombinant DNA reagent | pGBKT7 human PALLD N-term start-Ig2 (bp 212–1795; aa. 1–528; NM_001166108) | This paper | N/A | Cloning primers in *Supplementary file 1* |
| Recombinant DNA reagent | pGBKT7 mouse PALLD C-term Ig3-end (bp 1146–2309; aa 267–680; BC127081) | This paper | N/A | Cloning primers in *Supplementary file 1* |

*Appendix 1 Continued on next page*

*Appendix 1 Continued*

| Reagent type (species) or resource | Designation | Source or reference | Identifiers | Additional information |
|---|---|---|---|---|
| Recombinant DNA reagent | pGADT7-AD human CARP full-length | This paper | N/A | Cloning primers in *Supplementary file 1* |
| Recombinant DNA reagent | pGADT7-AD human FHOD1 (bp 3005–3607, aa. 965–1164; NM_013241.2) | This paper | N/A | Cloning primers in *Supplementary file 1* |
| Recombinant DNA reagent | pGADT7-AD human FHOD1 (bp3005-3268, aa. 965–1052; NM_013241.2) | This paper | N/A | Cloning primers in *Supplementary file 1* |
| Recombinant DNA reagent | pGADT7-AD human FHOD1 (bp 3152–3607, aa. 1014–1164; NM_013241.2) | This paper | N/A | Cloning primers in *Supplementary file 1* |
| Recombinant DNA reagent | pFN21A HaloTag CMV Flexi vector human PALLD full-length | This paper | N/A | Cloning primers in *Supplementary file 1* |
| Recombinant DNA reagent | pNLF1-N [CMV Hygro] mouse FHOD1 Y2H clone (bp3106-3711, aa. 997–1197); NM_013468.3 | This paper | N/A | Cloning primers in *Supplementary file 1* |
| Recombinant DNA reagent | pNLF1-C [CMV Hygro] mouse CARP (bp 64–1023; aa. 1–319; NM_013468.3) | This paper | N/A | Cloning primers in *Supplementary file 1* |
| Sequence-based reagent | qRT-PCR primers | This paper | N/A | *Supplementary file 1* |
| Commercial assay or kit | In-Fusion HD Cloning kit | Takara Bio | Cat# 639650 | |
| Commercial assay or kit | DC Protein Assay Kit II | Bio-Rad Laboratories | Cat# 5000112 | |
| Commercial assay or kit | Frozen-EZ Yeast Transformation II kit | Zymo Research | Cat# T2001 | |
| Commercial assay or kit | HaloTag NanoBRET 618 Ligand | Promega | Cat# G9801 | |
| Chemical compound, drug | Aureobasidin A | Takara Bio | Cat# 630499 | |
| Chemical compound, drug | X-α-Gal | Takara Bio | Cat# 630463 | |
| Chemical compound, drug | PureZOL RNA isolation reagent | Bio-Rad Laboratories | Cat# 7326890 | |
| Chemical compound, drug | High Capacity cDNA Reverse Transcription kit | Thermo Fisher Scientific | Cat# 4368814 | |
| Chemical compound, drug | SYBR Select Master Mix | Thermo Fisher Scientific | Cat# 4472903 | |
| Chemical compound, drug | Roche cOmplete Protease Inhibitor Cocktail | Thermo Fisher Scientific | Cat# 11697498001 | |
| Chemical compound, drug | Pierce Phosphatase Inhibitor Mini Tablets | Thermo Fisher Scientific | Cat# A32957 | |
| Chemical compound, drug | Immobilon Western Chemiluminescent HRP Substrate | Merck Life Science | Cat# WBKLS0500 | |

*Appendix 1 Continued on next page*

*Appendix 1 Continued*

| Reagent type (species) or resource | Designation | Source or reference | Identifiers | Additional information |
|---|---|---|---|---|
| Chemical compound, drug | Rhodamine phalloidin | Thermo Fisher Scientific | Cat# R415 | IF (1:100) |
| Chemical compound, drug | Wheat Germ Agglutinin, Alexa Fluor 594 Conjugate | Thermo Fisher Scientific | Cat# W11262 | IF (1:500) |
| Chemical compound, drug | VECTASHIELD Vibrance Antifade Mounting Medium with DAPI | D.B.A Italia Srl. | Cat# H-1800-10 | |
| Chemical compound, drug | Liberase Blendzyme | Roche Diagnostics | Cat# 11988468001 | |
| Chemical compound, drug | Fura-2, AM, cell permeant | Thermo Fisher Scientific | Cat# F1201 | |
| Software, algorithm | Prism, version 7.0 | GraphPad Software Inc. | https://www.graphpad.com/scientific-software/prism/ RRID:SCR_002798 | |
| Software, algorithm | Fiji (ImageJ) analysis software, version 2.0.0-rc-69/1.52p | National Institute of Health (NIH) | https://fiji.sc/ RRID:SCR_002285 | SparkMaster plugin used for $Ca^{2+}$ spark analysis |
| Software, algorithm | NT Affinity Analysis software, version 2.0.1334 | NanoTemper Technologies | N/A | |
| Software, algorithm | Ion Wizard, software, version 6.6.11 | IonOptix B.V. | N/A | |

