## [Editor Report]

This valuable work will be of interest to scientists who study cardiomyocyte homeostasis and contraction. Using solid methodology, this study is the first to assess the consequences of cardiomyocyte-specific knockout of Palladin, identifying a compensation mechanism that takes place when this gene is deleted in embryogenesis, but not in adulthood. In addition, this study identifies novel Palladin interactors, suggests a role for Palladin in the maintenance of intercalated disc structure, and assesses levels of Palladin expression in human samples.

---

## [Decision Letter]

**Decision letter after peer review:**

Thank you for submitting your article "Ablation of palladin in adult heart causes dilated cardiomyopathy associated with intercalated disc abnormalities" for consideration by *eLife*. Your article has been reviewed by 3 peer reviewers, one of whom is a member of our Board of Reviewing Editors, and the evaluation has been overseen by Didier Stainier as the Senior Editor. The following individuals involved in the review of your submission have agreed to reveal their identity: Jason R. Becker (Reviewer #2); W. Glen Pyle (Reviewer #3).

Essential revisions:

1) Authors propose that the DCM phenotype observed in the inducible PALLD cKOi hearts is a consequence of intercalated disc (ICD) defects. Please provide data assessing the integrity of the ICD in the constitutive PALLD cKO. If similar changes occur in these hearts, then this defect is not likely the cause of the phenotype observed in the inducible KO. Similarly, to ensure that the ICD phenotype is caused by the absence of PALLD rather than being a consequence of tamoxifen administration, please assess ICD integrity in Myh6:MerCreMer mice administered tamoxifen.

2) To ensure that apoptosis is not contributing to the phenotype observed, please repeat the TUNEL experiment at an earlier timepoint (4-week post-induction). In the current version of the manuscript, apoptosis was assessed 24-week post-induction, which is a rather late timepoint and might miss early events.

3) Please determine if CARP and FHOD1 localization are altered in PALLD cKO versus cKOi. This will improve the connection between the first section of the manuscript (identification of novel interactants) and the phenotypic characterization of conditional mutants, and might shed some light on why the two models diverge phenotypically.

4) As pointed out by all Reviewers, the human mRNA data alone does not provide enough evidence that PALLD might play a role in human DCM. It is particularly difficult to reconcile the fact that deletion of PALLD in mice causes DCM, whereas in human DCM patients the transcripts encoding this protein are upregulated. It is possible that PALLD protein levels are actually reduced in human DCM and the observed mRNA upregulation is part of a compensation mechanism. To test this, please provide measurements of PALLD protein level in human DCM samples.

*Reviewer #1 (Recommendations for the authors):*

1. The introduction has a long list of PALLD interactions with other proteins that is not necessary for the reader to understand the work here presented. Please shorten this text and simply mention that whereas multiple interactions have been previously identified for the proline-rich domains and last 3 Ig domains (as nicely summarized in Figure 1A), nothing is known regarding putative interactors in the N-terminal domain (encompassing the first 2 Ig domains).

2. Please alter Figure 1A to include not just the long isoform, but also the other relevant variants of PALLD. Please add also information regarding expected molecular weights. This will make it easier to understand in subsequent western blots what isoforms are expressed (and ablated in cKOs) in cardiomyocytes.

3. Is the signal of all antibodies used in Figure 1 specific? In other words, do these antibodies produce any signal when applied to tissues from PALLD KO/KO mice?

4. Based on the antigen information described in the text (whole protein, C-terminus, second proline-rich region), please add to Figure1 a diagram reporting what isoforms are recognized by each of the antibodies.

5. In Figure 1D, the nuclear labeling produced by antibody PALLD 622 seems to be enriched in specific foci. Is this a consistent observation, or a particular of this image? If this is consistent, do the authors know what sort of foci are these? Nucleoli?

6. The interactants and immunofluorescence data shown in Figure 1 are somehow disconnected from the rest of the manuscript. This could be improved if, for example, authors would perform immunofluorescence analyses on their cKOs to assess if the nuclear PALLD signal disappears in their cKO and icKO animals.

7. Lines 144-145: "we generated Palld floxed (Palldfl/fl) mice containing LoxP sites flanking exon 15". Please add here what is the expected consequence of ablation of exon 15 for the different isoforms of PALLD. What domain of the protein does exon 15 correspond to? After floxing out, is there an early termination codon leading to truncated variants?

8. Lanes 166-167: "in response to (…) TAC, no significant differences were found between cPKO and control mice as shown by echocardiography (Figure 2D and Figure 2—figure supplement 4) (…) although there was a tendency towards reduced contractile function in cPKO mice from 2 weeks after TAC". This description might cause some confusion, as the graphs do show significant differences (EF and FS) between cPKO and Cre- negative controls. Please revise the text to read "no significant differences were found between cPKO and Cre+ control mice".

9. Histological data in Figure 2E and respective quantification in 2F: in addition to the genotype, please add "4-week post TAC" to the label. Also, please highlight (with a box) in the low magnification whole heart image the area shown in the high magnification panels. The high magnification images show areas of interstitial myocardium, however, fibrosis post-TAC is frequently concentrated in the vicinity of large coronaries. Please provide also images and quantification of fibrosis in perivascular areas of control and cKO TAC hearts. Please include non-TAC hearts as control.

10. For increased clarity, in Figure 4A, please move genotype labels to the top, instead of the bottom position they currently occupy. On the left side of the panel, please add a label stating what sort of staining is being shown. Please highlight in the low magnification image the area shown in higher magnifications.

11. The discussion is rather long and has elements that belong in the Results section (for example describing tamoxifen injection into the constitutive KO mice).

12. Lines 460-461: "Based on the absence of a phenotype in PKO mice". Please correct to "Based on the absence of a phenotype in cPKO mice".

*Reviewer #2 (Recommendations for the authors):*

1. Are there changes in the PALLD interacting proteins depending on if PALLD is deleted in the embryonic versus adult period?

2. To better support the claim that ICD ultrastructural changes are contributing to DCM in adult PALLD deletion, it would be important to determine:

a. Do these same alterations in ICD ultrastructure occur in the mice with embryonic deletion of cardiomyocyte PALLD?

b. Are these ICD ultrastructure changes a non-specific result of ventricular dilation and failure? For example, do they occur in mice with DCM secondary to other causes (MI, chronic aortic banding, etc.)?

3. Since the ICD ultrastructure defects are postulated as the mechanism regulating DCM development in the adult PALLD deletion mice, it would be important to confirm that Myh6:MerCreMer mice administered tamoxifen do not develop similar ICD ultrastructural changes.

4. Apoptosis assessment at 2 or 4 weeks after tamoxifen administration would be helpful to better determine if cardiomyocyte apoptosis contributed to the 8 and 24 week DCM phenotype in the Myh6:MerCreMer/PALLD fl/fl mice.

5. Figure 4F – There appears to be a 4-fold reduction in Myh7 gene expression in Myh6:MerCreMer mice administered tamoxifen. Do the authors have an explanation for this result?

*Reviewer #3 (Recommendations for the authors):*

1. Is there a sufficient sample in the human tissue to measure protein levels or MYPN and PALLD to confirm (or refute) the mRNA analysis?

2. Identification of the novel binding partners is interesting but would be enhanced with data showing whether CARP and/or FHOD1 localization or levels are affected by the knockdown models. Without these links, the connection between Figure 1 and the rest of the paper is weakened.

3. It is not clear what the evidence is for the claim that PALLD is a candidate gene for cardiomyopathy. That changes in protein expression correlate with a disease phenotype (under certain conditions) is insufficient to offer this claim.

[Editors’ note: further revisions were suggested prior to acceptance, as described below.]

Thank you for resubmitting your work entitled "Ablation of palladin in adult heart causes dilated cardiomyopathy associated with intercalated disc abnormalities" for further consideration by *eLife*. Your revised article has been evaluated by Didier Stainier (Senior Editor) and a Reviewing Editor.

Despite considering that not all Reviewer's comments have been fully addressed, all three Reviewers recommend acceptance of the manuscript, provided that the Discussion section is altered to show awareness of study limitations. Please note that, as further detailed, additional corrections are needed in Figure 2 supplement 2.

Please submit the manuscript with the requested alterations highlighted so that we can proceed with formal acceptance.

Recommendations for the authors:

1. The new Figure 1, although helpful for the understanding of the paper, does not contain any results from original research. Having this in consideration, please make this a supplemental figure.

2. To conciliate the text with data presented and show awareness of study limitations please do the following alterations to the Discussion section:

2a. In line 385, after "unaffected", please include the following statement, or any combination of sentences conveying a similar message: "As measurements of contractility and ca^2+^ transients of isolated ventricular cardiomyocytes were performed at ca^2+^ concentrations that cannot be reached under physiological cardiac contraction, we cannot exclude that these observations might not fully recapitulate in vivo mechanisms".

2b. In line 430, after "dysfunction in cPKOi mice", please include the following statement, or any combination of sentences conveying a similar message: "Our data has shown that, in addition to the ICD, PALLD localizes to other cellular compartments, including the nucleus. Therefore, we cannot rule out the possibility that ultrastructural changes observed in cPKOi cardiomyocytes might be a consequence of LV dilation rather than the primary cause of the observed phenotype".

2c. New studies added to the revised manuscript showed absence of changes in MYPN and PALLD at the protein level in human patients. Therefore, please remove the following sentence (461-462): "Nevertheless, our results suggest the potential use of MYPN and PALLD as biomarkers".

2d. No disease-causing PALLD mutations are known, therefore correct lanes 469-470 to read "The reason why a putative link"

Shortly after submission of the revised manuscript, authors communicated to the editorial office a mistake in Figure 2 supplement 2. A corrected version of this figure has been submitted, however this needs further correction. The red channel image displayed for the middle panels does not match the merged image (it is still the same as in the original Figure 2 supplement 2). Please correct this.

---

## [Author Response]

Essential revisions:1) Authors propose that the DCM phenotype observed in the inducible PALLD cKOi hearts is a consequence of intercalated disc (ICD) defects. Please provide data assessing the integrity of the ICD in the constitutive PALLD cKO. If similar changes occur in these hearts, then this defect is not likely the cause of the phenotype observed in the inducible KO. Similarly, to ensure that the ICD phenotype is caused by the absence of PALLD rather than being a consequence of tamoxifen administration, please assess ICD integrity in Myh6:MerCreMer mice administered tamoxifen.

Since cPKO mice showed no pathological phenotype, we discontinued the line to reduce mouse costs and liberate space in the mouse facility. Therefore, unfortunately we don’t have the possibility to assess the integrity of the ICD in these mice. However, since ICD abnormalities are known to be associated with DCM, we consider it highly unlikely that cPKO mice would have ICD defects as observed in cPKOi mice. In fact, we are not aware of the existence of any mouse model with increased ICD fold amplitude and normal cardiac function.

In the previous Figure 5C and D, the groups named *Palld^fl/fl^*;TAM were in fact pooled results from MRM^+/0^ TAM and *Palld^fl/fl^*;TAM mice as we found no differences in ICD fold amplitude between the two control groups. We realize that we should have made this clear. We have now included the controls groups separately in the new Figure 6C and D. As it is clear from our measurements, TAM-injected Myh6:MerCreMer mice show no increase in ICD fold amplitude or other ICD abnormalities. It should be noted that Myh6:MerCreMer mice have been extensively used in the literature and that no adverse effects have been associated with administration of tamoxifen at a dose of 30 mg/kg/day for 5 days (Rouhi et al., *J Cardiovasc Aging* 2, 8, 2022).

2) To ensure that apoptosis is not contributing to the phenotype observed, please repeat the TUNEL experiment at an earlier timepoint (4-week post-induction). In the current version of the manuscript, apoptosis was assessed 24-week post-induction, which is a rather late timepoint and might miss early events.

As requested, we performed TUNEL staining on cPKOi (*Palld^fl/fl^*;MCM^+/-^) and control mice (*Palld^fl/fl^* and MCM^+/-^) 4 weeks after TAM induction. As shown in Author response image 1, no apoptosis was detected in cPKOi mice or any of the control mice 4 weeks after injection. Consequently, we have modified line 221 to: “No apoptosis was detected by TUNEL staining at 4 and 24 weeks after TAM induction”.

**Author response image 1. sa2fig1:** 

3) Please determine if CARP and FHOD1 localization are altered in PALLD cKO versus cKOi. This will improve the connection between the first section of the manuscript (identification of novel interactants) and the phenotypic characterization of conditional mutants, and might shed some light on why the two models diverge phenotypically.

The Reviewer may have missed that in Figure 6—figure supplement 2 (new Figure 7—figure supplement 2), we already provided immunofluorescence stainings, showing no alterations in the localization of CARP, FHOD1, and other known interaction partners of PALLD in cardiomyocytes (CMCs) from cPKOi *vs*. control mice. As we have discontinued the cPKO line, we don’t have the possibility to perform stainings on CMCs from cPKO mice. However, considering that we found no alterations in the localization of PALLD-interacting proteins in cPKOi mice, which have a pathological cardiac phenotype, we find it unlikely that CARP and FHOD1 would be mislocalized in CMCs of cPKO mice, which do not have any pathological cardiac phenotype.

To make it clearer that we performed immunofluorescence stainings for CARP, FHOD1 and other PALLD-interacting proteins, we modified the text in line 267-270 to:

“Immunofluorescence stainings of isolated adult CMCs showed no alterations in the localization of ICD proteins (Figure 7—figure supplement 1) and PALLD interacting proteins, including CARP, FHOD1, nebulette, SORBS2, MRTF-1, and cypher (Figure 7—figure supplement 2).”

4) As pointed out by all Reviewers, the human mRNA data alone does not provide enough evidence that PALLD might play a role in human DCM. It is particularly difficult to reconcile the fact that deletion of PALLD in mice causes DCM, whereas in human DCM patients the transcripts encoding this protein are upregulated. It is possible that PALLD protein levels are actually reduced in human DCM and the observed mRNA upregulation is part of a compensation mechanism. To test this, please provide measurements of PALLD protein level in human DCM samples.

We agree with the Reviewers that altered mRNA levels do not provide sufficient evidence for a role of PALLD in human DCM. As we didn’t have any tissue left of the original biopsies, we obtained new biopsies from 10 DCM and 7 ICM patients as well as 7 controls hearts on which we performed both qRT-PCR and Western blot analyses. qRT-PCR on the additional samples confirmed our previous results of increased transcript levels of the 200 kDa *PALLD* isoform, *MYPN*, and *ANKRD1* in DCM patients. In addition, we found a significantly reduction in overall transcript levels of *PALLD* in both DCM and ICM patients, which is consistent with the results of two RNA-Seq studies (Sweet et al., *BMC Genomics,* 19, 812, 2018; Yang et al., *Circulation,* 129, 1009-1021, 2014). The increased transcript level of the 200 kDa *PALLD* isoform despite the overall reduced transcript level of *PALLD* can be explained by the fact that the 200 kDa PALLD isoform is expressed at much lever than the 90 kDa PALLD isoform, which is the predominant isoform in the heart. By Western blot analyses, we found no changes in the protein expression of PALLD and MYPN in DCM and ICM patients *vs.* controls. On the other hand, the previously reported upregulation of CARP/Ankrd1 in DCM patients was confirmed (Kempton et al., *Heliyon*, 4:e00514, 2018; Nagueh et al., *Circulation,* 110, 155-162, 2014; Zolk et al., *Biochem Biophys Res Commun,* 293, 1377-1382, 2002). Thus, the transcriptional changes in *PALLD* and *MYPN* expression in human DCM are unlikely to affect cardiac function. Nevertheless, our findings suggest the potential use of *PALLD* and *MYPN* as biomarkers for DCM.

Reviewer #1 (Recommendations for the authors):1. The introduction has a long list of PALLD interactions with other proteins that is not necessary for the reader to understand the work here presented. Please shorten this text and simply mention that whereas multiple interactions have been previously identified for the proline-rich domains and last 3 Ig domains (as nicely summarized in Figure 1A), nothing is known regarding putative interactors in the N-terminal domain (encompassing the first 2 Ig domains).

To address the Reviewer’s comment, we removed the list of interactions with non-muscle proteins, which are not relevant for the role of PALLD in striated muscle. However, we prefer to include the list of PALLD-binding proteins in striated muscle as these are relevant for understanding the structural role of PALLD in the Z-line and ICD as well as its role in regulating the actin cytoskeleton. In our analysis of cPKOi mice, we performed Western blot and immunofluorescence staining for PALLD interacting proteins and we therefore find it important to mention them in the introduction.

2. Please alter Figure 1A to include not just the long isoform, but also the other relevant variants of PALLD. Please add also information regarding expected molecular weights. This will make it easier to understand in subsequent western blots what isoforms are expressed (and ablated in cKOs) in cardiomyocytes.

We agree that a figure showing the different PALLD isoforms will facilitate the understanding of the paper. As there are at least 10 PALLD isoforms it is not feasible to include all the isoforms in Figure 1A. Therefore, we have instead included a new Figure 1, illustrating all the currently known isoforms with predicted and apparent molecular weights, when known. In the new Figure 1—figure supplement 1, we have included a figure showing the exon structure of all known isoforms. This is particularly useful for understanding which isoforms are targeted by our approach.

3. Is the signal of all antibodies used in Figure 1 specific? In other words, do these antibodies produce any signal when applied to tissues from PALLD KO/KO mice?

All the used PALLD antibodies have been extensively characterized in different labs and are specific as shown by the absence of the targeted PALLD isoforms by Western blot analysis (Figure 3B and 4B).

Due to extensive isoform diversity, it wasn’t possible to target all PALLD isoforms and in cPKO mice both an N-terminal and a C-terminal isoform not targeted by our approach are expressed (see the new Figure 1—figure supplement 1). Unfortunately, a C-terminal isoform with a start codon 3’ of the exon that we targeted was discovered after the mice had been generated. For that reason, the PALLD antibodies also stain cPKO tissue and it is therefore difficult to distinguish cPKO mice from control mice based on immunofluorescence stainings. Thus, to avoid confusion, we have not included pictures of KO tissue stained with PALLD antibodies.

4. Based on the antigen information described in the text (whole protein, C-terminus, second proline-rich region), please add to Figure1 a diagram reporting what isoforms are recognized by each of the antibodies.

The epitopes against which the PALLD (Proteintech) and 4D10 antibodies have been raised are indicated in the new Figure 1. Unfortunately, although the PALLD 622 antibody (and the PALLD 621 antibody raised against the same epitopes) has been extensively used in the literature, it has not been possible to find out the exact epitopes to which it was raised and for this reason we have not included the PALLD 622 antibody in the figure. From personal communication with Prof. Carol Otey, in whose lab the PALLD 622 antibody was generated many years ago, the antibody was raised against both N- and C-terminal epitopes and should thus recognize all isoforms.

5. In Figure 1D, the nuclear labeling produced by antibody PALLD 622 seems to be enriched in specific foci. Is this a consistent observation, or a particular of this image? If this is consistent, do the authors know what sort of foci are these? Nucleoli?

We thank the Reviewer for pointing this out. Both the PALLD 622 and Proteintech antibodies showed the absence of PALLD from nuclear regions strongly stained with DAPI, suggesting its localization in the transcriptionally active regions of the nucleus. Furthermore, with the PALLD 622 antibody we often observed a punctuate staining pattern, which appears to correspond to nucleoli. To point this out we have added the following sentence:

Line 121: “PALLD was found to be absent from nuclear regions strongly stained with DAPI, corresponding to heterochromatin, suggesting the localization of PALLD in the transcriptionally active euchromatin regions of the nucleus. Additionally, stronger PALLD staining in a punctuate pattern was often observed in regions with absent DAPI staining, likely corresponding to nucleoli”.

6. The interactants and immunofluorescence data shown in Figure 1 are somehow disconnected from the rest of the manuscript. This could be improved if, for example, authors would perform immunofluorescence analyses on their cKOs to assess if the nuclear PALLD signal disappears in their cKO and icKO animals.

As mentioned above, unfortunately, due to extensive isoform diversity it is not possible to target all PALLD isoforms, so there are both an N-terminal and a C-terminal isoform not targeted by our approach (see the new Figure 1—figure supplement 1). For that reason PALLD staining is also observed in cPKO and cPKOi mice and to avoid confusion we therefore didn’t include immunofluorescence stainings on KO animals.

7. Lines 144-145: "we generated Palld floxed (Palldfl/fl) mice containing LoxP sites flanking exon 15". Please add here what is the expected consequence of ablation of exon 15 for the different isoforms of PALLD. What domain of the protein does exon 15 correspond to? After floxing out, is there an early termination codon leading to truncated variants?

To address this, we have now included Figure 1—figure supplement 1, showing the exon structure of all known PALLD isoforms. The floxed exon is indicated with yellow and corresponds to a region between the second proline-rich region and the third Ig domain. The length of the floxed exon is not evenly divisible by 3 and Cre-mediated recombination thus causes a frameshift and consequently a premature stop codon. This results in RNA instability and consequent RNA degradation by nonsense-mediated decay. Thus, the location of the floxed exon is not important as all isoforms containing the exon get degraded.

We initially referred to the targeted exon as exon 15 according to the nomenclature of Wang and Moser (Wang and Moser, 2008). However, while preparing the figure, we realized that exon 12 had been counted as two exons by Wang and Moser and for that reason we now refer to the floxed exon as exon 14. As evident from the figure, the last two isoforms shown are not targeted by our approach.

8. Lanes 166-167: "in response to (…) TAC, no significant differences were found between cPKO and control mice as shown by echocardiography (Figure 2D and Figure 2—figure supplement 4) (…) although there was a tendency towards reduced contractile function in cPKO mice from 2 weeks after TAC". This description might cause some confusion, as the graphs do show significant differences (EF and FS) between cPKO and Cre- negative controls. Please revise the text to read "no significant differences were found between cPKO and Cre+ control mice".

The text has been revised as requested in line 169 of the revised manuscript. Additionally, in the discussion “although cPKO mice showed a tendency towards reduced systolic function compared to control mice” has been removed from the sentence in line 379, so it now reads:

“Unexpectedly, cPKO mice exhibited no pathological cardiac phenotype at basal levels and showed a normal hypertrophic response to TAC”.

9. Histological data in Figure 2E and respective quantification in 2F: in addition to the genotype, please add "4-week post TAC" to the label. Also, please highlight (with a box) in the low magnification whole heart image the area shown in the high magnification panels. The high magnification images show areas of interstitial myocardium, however, fibrosis post-TAC is frequently concentrated in the vicinity of large coronaries. Please provide also images and quantification of fibrosis in perivascular areas of control and cKO TAC hearts. Please include non-TAC hearts as control.

As requested by the Reviewer, we have included hearts from mice subjected to SHAM as controls, indicated the location of high-magnification areas with a box, and modified the labels (new Figure 3E). In addition, we have included pictures of perivascular areas and performed the quantification of both interstitial and perivascular areas. No significant difference in fibrosis was found between cPKO and control hearts.

10. For increased clarity, in Figure 4A, please move genotype labels to the top, instead of the bottom position they currently occupy. On the left side of the panel, please add a label stating what sort of staining is being shown. Please highlight in the low magnification image the area shown in higher magnifications.

The figure (now Figure 5A) has been modified as requested by the Reviewer. In addition, pictures of perivascular areas have been included and quantification of both interstitial and perivascular areas has been performed. Both interstitial and perivascular fibrosis were significantly increased in cPKOi hearts.

11. The discussion is rather long and has elements that belong in the Results section (for example describing tamoxifen injection into the constitutive KO mice).

We agree that the tamoxifen injection of cPKO mice belongs to the result section. This has now been moved to line 195-200 of the result section and the results are presented in Figure 4—figure supplement 1.

Line 200-205: “To rule out the possibility that the phenotype of cPKOi mice is due to the toxic effect of TAM, we injected cPKO mice with TAM and evaluated cardiac function by echocardiography 8 weeks after TAM injection (Figure 4—figure supplement 1). No pathological cardiac phenotype was observed after TAM injection, suggesting that early ablation of PALLD allows for the activation of compensatory mechanisms.”

We are not aware of other elements of the discussion belonging to the result section.

12. Lines 460-461: "Based on the absence of a phenotype in PKO mice". Please correct to "Based on the absence of a phenotype in cPKO mice".

Thanks for noticing this. This has now been corrected.

Reviewer #2 (Recommendations for the authors):1. Are there changes in the PALLD interacting proteins depending on if PALLD is deleted in the embryonic versus adult period?

As we didn’t find any changes in the expression or localization of PALLD interacting proteins in cPKOi mice, which have a pathological cardiac phenotype, we didn’t check for changes in cPKO mice, which have no pathological phenotype. Unfortunately, the cPKO line has been discontinued as a result of the absent pathological phenotype and it is therefore not possible to check. Considering that we found no alterations in localization of PALLD-interacting proteins in cPKOi mice, which have a pathological cardiac phenotype, we find it unlikely that PALLD interacting proteins would be mislocalized cPKO mice.

2. To better support the claim that ICD ultrastructural changes are contributing to DCM in adult PALLD deletion, it would be important to determine:a. Do these same alterations in ICD ultrastructure occur in the mice with embryonic deletion of cardiomyocyte PALLD?

As explained above, since cPKO mice showed no pathological phenotype, we discontinued the line to reduce mouse costs and liberate space in the mouse facility. Therefore, unfortunately we don’t have the possibility to assess the integrity of the ICD in these mice. However, since ICD abnormalities are known to be associated with DCM, we consider it highly unlikely that cPKO mice would have ICD defects as observed in cPKOi mice. In fact, we are not aware of the existence of any mouse model with increased ICD fold amplitude and normal cardiac function.

b. Are these ICD ultrastructure changes a non-specific result of ventricular dilation and failure? For example, do they occur in mice with DCM secondary to other causes (MI, chronic aortic banding, etc.)?

ICD widening and been observed in various mouse models of DCM and ICD disorganization has been associated with human DCM (Wilson et al., *Cell Mol Life Sci, 71*, 165-181, 2014; Ito et al., Scientific reports 11, 11852, 2021; Ortega et al., PLoS One 12; e0185062, 2017). Also, chronic TAC has been associated with ICD widening (Zankov et al., Scientific Reports 7, 39335, 2017). However, it is unclear whether ICD disorganization is a cause or a result of DCM. We previously reported that MYPN KO mice subjected to 4 weeks of TAC develop severe DCM associated with increased ICD fold amplitude (Filomena et al., *eLife* 10:e58313, 2021). However, although MYPN subjected to TAC develop a much more severe DCM phenotype than cPKOi mice, the ICM abnormalities are more severe in cPKOi mice. Based on this, the ICD abnormalities in cPKOi mice appear to be more than just a secondary result of the DCM phenotype and likely related to the association of PALLD with the ICD.

3. Since the ICD ultrastructure defects are postulated as the mechanism regulating DCM development in the adult PALLD deletion mice, it would be important to confirm that Myh6:MerCreMer mice administered tamoxifen do not develop similar ICD ultrastructural changes.

In the previous Figure 5C and D, the groups named *Palld^fl/fl^*;TAM were in fact pooled results from MRM^+/0^ TAM and *Palld^fl/fl^*;TAM mice as we found no differences in ICD fold amplitude between the two control groups. We realize that we should have made this clear. We have now included the controls groups separately in the new Figure 6C and D. As it is clear form our measurements, TAM-injected Myh6:MerCreMer mice show no increase in ICD fold amplitude or other ICD abnormalities. It should be noted that Myh6:MerCreMer mice have been extensively used in the literature and that no adverse effects have been associated with administration of tamoxifen at a dose of 30 mg/kg/day for 5 days (Rouhi et al., *J Cardiovasc Aging* 2, 8, 2022).

4. Apoptosis assessment at 2 or 4 weeks after tamoxifen administration would be helpful to better determine if cardiomyocyte apoptosis contributed to the 8 and 24 week DCM phenotype in the Myh6:MerCreMer/PALLD fl/fl mice.

Based on the literature there is no reason to think that administration of tamoxifen at a dose of 30 mg/kg/day for 5 days to Myh6:MerCreMer mice would result in apoptosis. As requested, we have performed TUNEL staining on cPKOi (*Palld^fl/fl^*;MCM^+/-^) and control mice (*Palld^fl/fl^* and MCM^+/-^) 4 weeks after TAM induction. As shown in Author response image 1, no apoptosis was detected in cPKOi or any of the control mice 4 weeks after injection. Consequently, we have modified line 221 to: “No apoptosis was detected by TUNEL staining at 4 and 24 weeks after TAM induction”.

5. Figure 4F – There appears to be a 4-fold reduction in Myh7 gene expression in Myh6:MerCreMer mice administered tamoxifen. Do the authors have an explanation for this result?

In a recent qRT-PCR study, TAM induction of Myh6:MerCreMer mice at a dose of 30 mg/kg/day for 5 days was found to cause transient changes in CMC gene expression 2 weeks after TAM injection without affecting cardiac function, myocardial fibrosis, apoptosis, or induction of double-stranded DNA breaks (Rouhi et al., *J Cardiovasc Aging* 2, 8, 2022). Consistent with the downregulation of *Myh7* in our study, *Myh7* expression was reported to be 1.9-fold reduced in CMCs 2 weeks after TAM induction. Thus, the downregulation of Myh7 is likely an effect of the TAM induction of Myh6:MerCreMer mice.

Reviewer #3 (Recommendations for the authors):1. Is there a sufficient sample in the human tissue to measure protein levels or MYPN and PALLD to confirm (or refute) the mRNA analysis?

We didn’t have sufficient human tissue tissues left for measuring protein of MYPN and PALLD. However, we were able to obtain new biopsies from 10 DCM and 7 ICM patients as well as 7 controls hearts on which we performed both qRT-PCR and Western blot analyses. qRT-PCR on the additional samples confirmed our previous results of increased transcript levels of the 200 kDa *PALLD* isoform, *MYPN*, and *ANKRD1* in DCM patients. In addition, we found a significantly reduction in overall transcript levels of *PALLD* in both DCM and ICM patients, which is consistent with the results of two RNA-Seq studies (Sweet et al., *BMC Genomics,* 19, 812, 2018; Yang et al., *Circulation,* 129, 1009-1021, 2014). The increased transcript level of the 200 kDa *PALLD* isoform despite the overall reduced transcript level of *PALLD* can be explained by the fact that the 200 kDa PALLD isoform is expressed at much lever than the 90 kDa PALLD isoform, which is the predominant isoform in the heart. By Western blot analyses, we found no changes in the protein expression of PALLD and MYPN in DCM and ICM patients *vs.* controls. On the other hand, the previously reported upregulation of CARP/Ankrd1 in DCM patients was confirmed (Kempton et al., *Heliyon*, 4:e00514, 2018; Nagueh et al., *Circulation,* 110, 155-162, 2014; Zolk et al., *Biochem Biophys Res Commun,* 293, 1377-1382, 2002). Thus, the transcriptional changes in *PALLD* and *MYPN* expression in human DCM are unlikely to affect cardiac function.

2. Identification of the novel binding partners is interesting but would be enhanced with data showing whether CARP and/or FHOD1 localization or levels are affected by the knockdown models. Without these links, the connection between Figure 1 and the rest of the paper is weakened.

The Reviewer may have missed that in Figure 6—figure supplement 2 (new Figure 7—figure supplement 2), we already provided immunofluorescence stainings, showing no alterations in the localization of CARP, FHOD1, and other known interaction partners of PALLD in CMCs from cPKOi *vs*. control mice. As we have discontinued the cPKO line, we don’t have the possibility to perform stainings on CMCs from cPKO mice. However, considering that we found no alterations in the localization of PALLD-interacting proteins in cPKOi mice, which have a pathological cardiac phenotype, we find it unlikely that CARP and FHOD1 would be mislocalized in cPKO CMCs.

To make it clearer that we performed immunofluorescence stainings for CARP, FHOD1 and other PALLD-interacting proteins, we modified the text in line 267-269 to:

“Immunofluorescence stainings of isolated adult CMCs showed no alterations in the localization of ICD proteins (Figure 7—figure supplement 1) and PALLD interacting proteins, including CARP, FHOD1, nebulette, SORBS2, MRTF-1, and cypher (Figure 7—figure supplement 2).

3. It is not clear what the evidence is for the claim that PALLD is a candidate gene for cardiomyopathy. That changes in protein expression correlate with a disease phenotype (under certain conditions) is insufficient to offer this claim.

This claim was made based on the phenotype of the cPKOi mice, not the fact that *PALLD* transcript levels are changed in DCM patients. In the paper we demonstrated that PALLD plays a role in the heart. Like for its homologue MYPN, it is possible that dominant PALLD mutations could be associated with cardiac disease, while PALLD ablation would likely lead to embryonic lethality due to the essential role of PALLD in other tissues and cell types. MYPN KO mice show only mild DCM, while human *MYPN* mutations with dominant negative effects have been shown to cause dilated, hypertrophic, and restrictive cardiomyopathy (Filomena et al., *eLife* 10: e58313, 2021). We discuss this in the last section of the discussion.

[Editors’ note: further revisions were suggested prior to acceptance, as described below.]

Recommendations for the authors:1. The new Figure 1, although helpful for the understanding of the paper, does not contain any results from original research. Having this in consideration, please make this a supplemental figure.

We have now included Figure 1 and Figure 1—figure supplement 1 as new Figure 1—figure supplement 1 and Figure 1—figure supplement 2. Consequently, all figure numbers have been changed.

2. To conciliate the text with data presented and show awareness of study limitations please do the following alterations to the Discussion section:2a. In line 385, after "unaffected", please include the following statement, or any combination of sentences conveying a similar message: "As measurements of contractility and ca^2+^ transients of isolated ventricular cardiomyocytes were performed at ca^2+^ concentrations that cannot be reached under physiological cardiac contraction, we cannot exclude that these observations might not fully recapitulate in vivo mechanisms".

The Ionoptix studies on isolated cardiomyocytes were performed in the presence of a physiological amount of CaCl_2_ (1.0 mM). It was the experiments on myofibril preparations that were performed at non-physiological ca^2+^ concentrations. We have therefore modified the suggested sentence to the following:

“As biomechanical properties of myofibril preparations from the LV were measured at ca^2+^ concentrations that cannot be reached under the physiological conditions of cardiac contraction, we cannot exclude that our observations might not fully recapitulate in vivo mechanisms”.

2b. In line 430, after "dysfunction in cPKOi mice", please include the following statement, or any combination of sentences conveying a similar message: "Our data has shown that, in addition to the ICD, PALLD localizes to other cellular compartments, including the nucleus. Therefore, we cannot rule out the possibility that ultrastructural changes observed in cPKOi cardiomyocytes might be a consequence of LV dilation rather than the primary cause of the observed phenotype".2c. New studies added to the revised manuscript showed absence of changes in MYPN and PALLD at the protein level in human patients. Therefore, please remove the following sentence (461-462): "Nevertheless, our results suggest the potential use of MYPN and PALLD as biomarkers".2d. No disease-causing PALLD mutations are known, therefore correct lanes 469-470 to read "The reason why a putative link"

The requested changes 2b, 2c, 2d to the discussion have been made as suggested.

Shortly after submission of the revised manuscript, authors communicated to the editorial office a mistake in Figure 2 supplement 2. A corrected version of this figure has been submitted, however this needs further correction. The red channel image displayed for the middle panels does not match the merged image (it is still the same as in the original Figure 2 supplement 2). Please correct this.

We apologize for this mistake. The figure has now been corrected.